Methods

# High-quality single-cell transcriptomics from ovarian histological sections during folliculogenesis

Hiroki Ikeda[1], Shintaro Miyao[1], So Nagaoka[1], Tomoya Takashima[1], Sze-Ming Law[1], Takuya Yamamoto[2,3,4], Kazuki Kurimoto[1,5]

High-quality, straightforward single-cell RNA sequencing (RNA-seq) with spatial resolution remains challenging. Here, we developed DRaqL (direct RNA recovery and quenching for laser capture microdissection), an experimental approach for efficient cell lysis of tissue sections, directly applicable to cDNA amplification. Single-cell RNA-seq combined with DRaqL allowed transcriptomic profiling from alcohol-fixed sections with efficiency comparable with that of profiling from freshly dissociated cells, together with effective exon–exon junction profiling. The combination of DRaqL with protease treatment enabled robust and efficient single-cell transcriptome analysis from formalin-fixed tissue sections. Applying this method to mouse ovarian sections, we were able to predict the transcriptome of oocytes by their size and identified an anomaly in the size–transcriptome relationship relevant to growth retardation of oocytes, in addition to detecting oocyte-specific splice isoforms. Furthermore, we identified differentially expressed genes in granulosa cells in association with their proximity to the oocytes, suggesting distinct epigenetic regulations and cell-cycle activities governing the germ–soma relationship. Thus, DRaqL is a versatile, efficient approach for high-quality single-cell RNA-seq from tissue sections, thereby revealing histological heterogeneity in folliculogenic transcriptome.

## Introduction

Single-cell RNA sequencing (RNA-seq) was first achieved by using a quantitative cDNA amplification method and applied to mouse oocytes (Kurimoto et al, 2006; Tang et al, 2009). It has since provided unprecedented opportunities for the study of cellular differentiations, states, and diseases in various biological fields, including developmental biology, stem cell biology, and reproductive medicine. Although many high-throughput single-cell RNA-seq methods have been developed (see Svensson et al [2018] for review), they typically lose histological information during cell dissociation from tissues. To preserve the histological information in transcriptome analyses, various in situ spatial transcriptomic methods that cover whole-tissue sections have been developed (see Liao et al [2021] for review). These single-cell and spatial transcriptomics are extremely high throughput, rely on unique molecular identifiers or hybridization probes, and output relatively low transcriptomic contents with a low signal-to-noise ratio, in comparison with conventional deep RNA sequencing; usually, for example, the detectable number of genes ranges from hundreds to a few thousand, and exon–exon junctions and sequence variants are not identified (Waylen et al, 2020; Liao et al, 2021). As a result, the developed transcriptomics is optimal for identifying cell types in a large cell population and/or spatially annotating them, but are likely suboptimal for in-depth analyses of individual cells in tissues. For example, oogenesis undergoes quality control of oocytes during folliculogenesis accompanied by intimate interactions between oocytes and surrounding granulosa cells, and thus, understanding this process would require high-quality single-cell transcriptomics tightly linked with histology (Zhang et al, 2018).

On the other hand, comprehensive, unbiased transcriptomics has been achieved and widely applied for arbitrarily targeted regions of interest (ROIs) isolated from tissue sections with laser capture microdissection (LCM) (Espina et al, 2006). LCM-based transcriptomics targets in situ cells/regions for deep RNA sequencing, but it has conventionally been used for bulk tissue fragments. On the other hand, several methods have been developed for LCM-based single-cell and/or low-input RNA-seq (Nichterwitz et al, 2016; Chen et al, 2017; Foley et al, 2019; Perez et al, 2021). Thus, this approach has advantages for use in the performance of unbiased, comprehensive transcriptomics in histologically identifiable small numbers of cells, including for the detection of exon junctions, and would be expected to provide information complementary to that of the currently available high-throughput single-cell RNA-seq and spatial transcriptomics (Liao et al, 2021). In addition, in situations calling for the analysis of cells/

---

[1]Department of Embryology, School of Medicine, Nara Medical University, Kashihara, Japan   [2]Center for iPS Cell Research and Application (CiRA), Kyoto University, Kyoto, Japan   [3]Institute for the Advanced Study of Human Biology (WPI-ASHBi), Kyoto University, Kyoto, Japan   [4]Medical-risk Avoidance based on iPS Cells Team, RIKEN Center for Advanced Intelligence Project (AIP), Kyoto, Japan   [5]Advanced Medical Research Center, Nara Medical University, Kashihara, Japan

Correspondence: kurimoto@naramed-u.ac.jp

ROI across many sections, the targeted isolation strategy using LCM would be particularly cost-effective.

In the earliest version of LCM-based single-cell RNA-seq, the cDNA amplification method for Smart-seq2 (Picelli et al, 2013) was directly applied to alcohol-fixed sections, with cell lysis using a non-denaturing detergent (Triton X-100) (Nichterwitz et al, 2016). Non-denaturing detergents are most frequently used for the lysis of freshly dissociated single cells, and such detergents can also be used in the subsequent enzymatic reactions for cDNA amplification in the same sampling tubes, a critical attribute for the success of low-input analyses such as analyses of single cells. On the other hand, the lysis efficiency of cells from sections with non-denaturing detergents has been controversial, which led Chen et al (2017) to propose a strategy that subjects a small number of cells in tissue sections to complete cell lysis under a denaturing condition followed by RNA purification with ethanol precipitation. Although efficient, high-quality RNA recovery is critical for quantitative transcriptome analysis of single cells, and the laborious procedures involved in the RNA purification might limit the practical utility of such an approach (Le et al, 2015; Ghimire et al, 2021).

Moreover, formalin-fixed sections remain a challenge for high-quality, comprehensive single-cell RNA-seq, although formalin fixation achieves good tissue preservation and is widely used in histology (Titford & Horenstein, 2005; Paavilainen et al, 2010). Although a two-way RNA-seq method was developed for both alcohol- and formalin-fixed sections (Foley et al, 2019), in principal, it relied on an additional RNA hydrolysis for cell lysis and a short elongation time for cDNA amplification, thereby excluding transcript information other than the 3′-ends, and potentially compromising the sensitivity as well. Similarly, current single-cell transcriptomics often relies on the detection of 3′-ends or targeting probes, and frequently neglects additional sequence information such as exon–exon junctions.

Thus, an efficient, versatile cDNA amplification method for alcohol- and formalin-fixed tissue sections without RNA purification would enable comprehensive and robust in situ single-cell transcriptomics for histologically targeted cells-of-interest in a less labor-intensive manner. In this study, we developed cDNA amplification methods combined with an efficient cell lysis strategy for tissue sections that uses a denaturing detergent for lysis, followed by quenching of the denaturing effect with an excess amount of a non-denaturing detergent (direct RNA recovery and quenching for LCM [DRaqL]). The versatility of DRaqL was demonstrated by using it in combination with three different cDNA amplification protocols: SC3-seq (Kurimoto et al, 2006; Nakamura et al, 2015), Smart-seq2 (Picelli et al, 2013), and the protocol in the SMART-Seq v4 3′DE kit, which is a commercially available Smart-seq2-based kit that is compatible with multiplex cDNA library preparation and allows improved throughput (Takara Bio). The DRaqL-combined methods allowed efficient transcriptome profiling and exon–exon junction analyses of single cells isolated from alcohol-fixed sections with LCM. The quality of the analysis was comparable with those of freshly dissociated single cells. Furthermore, when combined with protease treatment, DRaqL was successfully applied to tissue sections strongly fixed with formalin (10%, 24 h at room temperature), enabling reliable single-cell RNA-seq from formalin-fixed sections.

By applying this method to mouse ovarian sections, we revealed a transcriptomic continuum of growing oocytes and detected splice isoforms important in oogenesis. We constructed a statistical model of the transcriptome of oocytes based on their size, and, by examining deviations from the model, we found heterogeneity of the size–transcriptome relationship in oocytes, relevant to growth retardation. Moreover, we revealed genes that were differentially expressed in granulosa cells in association with their histological parameters. We thus established a versatile, effective single-cell cDNA amplification strategy for high-quality RNA-seq from alcohol- and formalin-fixed tissue sections, and revealed histology-associated transcriptomic heterogeneity in mouse folliculogenesis.

# Results

## Experimental system for quantitative examination of cDNA amplification from sections

First, we sought a method to efficiently amplify single-cell cDNAs from alcohol-fixed sections. To circumvent cellular heterogeneity in tissues, we set up a system to evaluate the cDNA amplification of single cells isolated with LCM from the alcohol-fixed, stained sections of frozen-cell blocks composed of homogeneous cultured cells—namely, embryonic stem cells under a 2i-LIF condition (2i-LIF mouse embryonic stem cells [mESCs]) (Ying et al, 2008; Marks et al, 2012) (Fig 1A and B). As a gold standard, fresh single cells from the same culture batches were also dissociated and isolated. We amplified cDNAs of these single cells by means of the amplification protocol used in the SC3-seq method (Kurimoto et al, 2006, 2008; Nakamura et al, 2015), and compared their gene expression levels using real-time PCR.

We first evaluated embedding media and found that the gene expression profiles were more compromised by optimal cutting temperature compound, a widely used embedding medium, than by using 10% polyvinyl alcohol (PVA) (Fig S1A and B). Thus, we decided to use 10% PVA for embedding.

## Efficient cDNA amplification with DRaqL-SC3-seq

Next, we evaluated the cDNA amplification using a non-denaturing detergent, Triton X-100 (0.63%; SC3-seq cDNA amplification [Triton X-100]), for single cells isolated from sections of the cell blocks. In agreement with the previous study (Chen et al, 2017), the cells isolated from the cell-block sections showed significantly reduced expression levels of *Arbp*, a highly expressed housekeeping gene, compared with the freshly dissociated cells (Fig 1C).

We therefore hypothesized that the use of denaturing detergents would improve the cell lysis efficiency, and that subsequent enzymatic reactions would be allowed by quenching the denaturing effect with the addition of an excess amount of non-denaturing detergents (Fig 1B). We evaluated the quenching effect with cDNA amplification from the single-cell equivalent amount (10 pg) of total RNA, and found that the denaturing detergents sodium deoxycholate (SDc) (≤0.63%) and SDS (≤0.25%) were efficiently quenched by the addition of 7.6% Triton X-100 (Fig 1D and E). In addition,

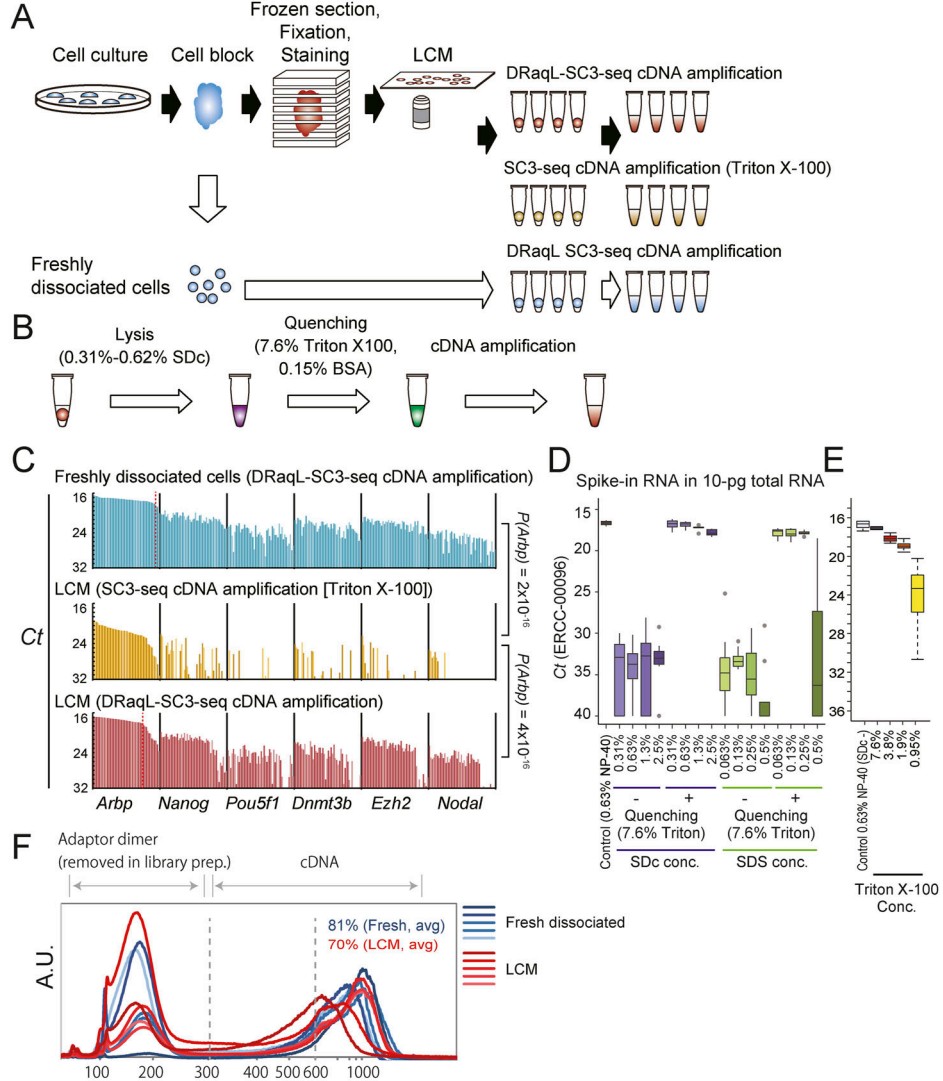

Figure 1. Single-cell DRaqL–SC3-seq cDNA amplification from cell blocks.
**(A)** Schematic representation of the development of the DRaqL–SC3-seq cDNA amplification method. **(B)** Schematic representation of DRaqL-combined cDNA amplification. **(C)** Real-time PCR analysis of cDNAs of freshly dissociated single mESCs (top), single cells isolated from alcohol-fixed sections of cell blocks, lysed using Triton X-100 (middle) and DRaqL (bottom) (n = 48 each). cDNAs were amplified with the indicated methods. $P$-values by a $t$ test for $Arbp$ $C_t$ values are also shown. **(D)** Evaluation of the SC3-seq cDNA amplification efficiency from 10-pg RNA of mESCs with different concentrations of SDc and SDS, with and without quenching by 7.6% Triton X-100. $C_t$ values of spike-in ERCC-00096 RNA (~3,000 copies) are represented with box plots (n = 8 each). **(E)** Evaluation of cDNA amplification efficiency with different concentrations of Triton X-100 for quenching of 0.63% SDc (n = 4 each). **(F)** Electropherograms of cDNAs analyzed using Bioanalyzer (Agilent Technologies). The adapter dimers (<300 bps) are removed during library preparation.

efficient quenching required BSA at a concentration up to 0.15% (Fig S1C). Thus, we decided to use 0.31–0.63% SDc for the cell lysis, and 7.6% Triton X-100 and 0.15% BSA for the quenching, and we termed the lysis method DRaqL. When we combined DRaqL with the SC3-seq cDNA amplification, we used 0.63% SDc for cell lysis.

Next, we evaluated the DRaqL–SC3-seq cDNA amplification using single cells isolated from the alcohol-fixed sections of the cell blocks (48 cells each). As shown in Figs 1C and S1D, the amplification efficiency was significantly improved compared with the efficiency of SC3-seq cDNA amplification (Triton X-100) ($P = 4 \times 10^{-16}$), and was similar to the amplification efficiency from freshly dissociated cells, albeit with a ~19% reduction in the success rate (Fig S1E).

We also found that the electropherograms of amplified cDNAs were similar between these cells, with only a small size reduction in the cell-block cells (~81% and ~70% of cDNA >600 bps, respectively) (Fig 1F). Thus, the DRaq-SC3-seq cDNA amplification is a useful single-cell cDNA amplification method from alcohol-fixed sections.

## Examination of transcriptome profiling with DRaqL–SC3-seq from alcohol-fixed sections

Next, we examined the amplified cDNA in a genome-wide manner, using the 3′-sequencing method, SC3-seq (Nakamura et al, 2015, 2016) (Fig 2A). We found that the freshly dissociated cells and cell-block cells showed similar numbers of detected genes (10,730 and 9,886 genes on average, respectively) (Fig 2B and C). The expression levels of spike-in RNAs (ERCCs) were also essentially the same in both types of cells (Fig S2A and B). Scatterplots of gene expression levels in single cells showed no large difference between these types of cells (Fig 2D). Consistent with the above findings, principal component analysis (PCA) showed that about 40% of the 97% confidence interval ellipse areas were overlapped between the freshly dissociated cells and cell-block cells (Fig 2E). In addition, the average gene expression patterns were similar between the freshly dissociated cells and cell-block cells, with only small systematic errors, as described below (Fig 2F). These results demonstrate that DRaqL–SC3-seq allows a high-

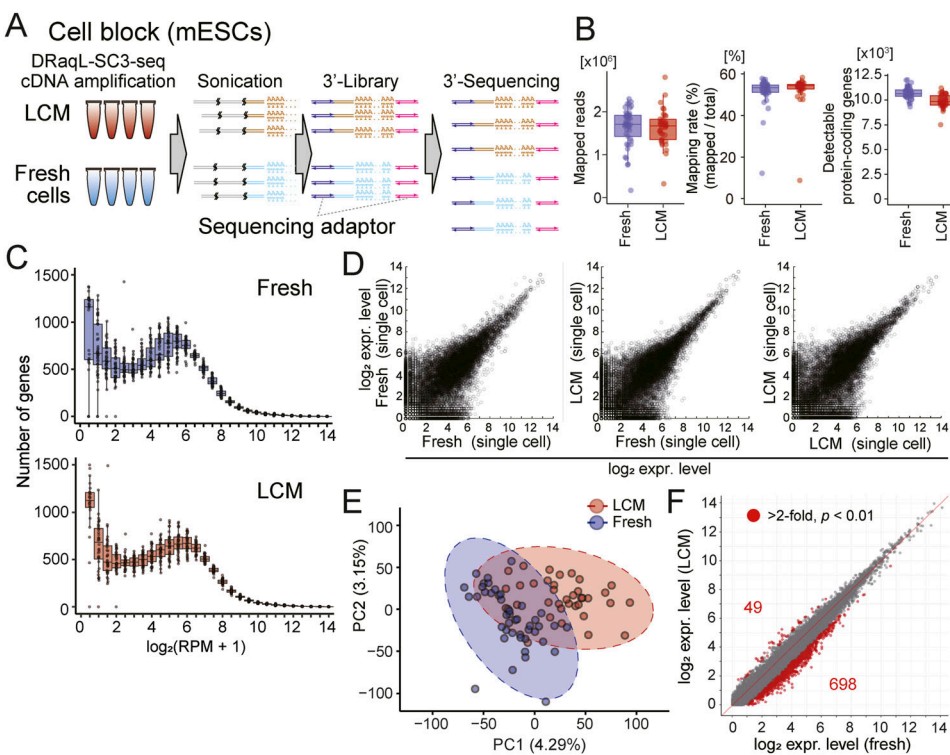

**Figure 2. Single-cell transcriptome analysis with DRaqL-SC3-seq.**
**(A)** Schematic representation of the 3′-sequencing with DRaqL-SC3-seq. **(B)** Box plots showing the numbers of mapped reads (left), mapping rates (middle), and detectable protein-coding genes (>0 mapped reads) (right). Freshly dissociated single cells (fresh) and single cells from alcohol-fixed cell-block sections are shown (LCM). **(C)** Frequency plots of gene expression levels in freshly dissociated single cells (fresh) (top), and single cells from alcohol-fixed cell-block sections (LCM) (bottom). **(D)** Representative scatterplots of gene expression levels between freshly dissociated single cells (fresh) and single cells isolated from alcohol-fixed cell-block sections (LCM). **(E)** PCA of freshly dissociated single cells (fresh) and single cells isolated from alcohol-fixed cell blocks (LCM). 97% confidence interval ellipses are represented with dashed lines. **(F)** Scatterplots of the averaged log$_2$ (RPM+1) values between freshly dissociated single cells and single cells isolated from alcohol-fixed cell-block sections. Differentially expressed genes (log$_2$ difference >1, and $P < 0.01$ by $t$ test) are indicated with red circles. Source data are available for this figure.

quality single-cell transcriptome analysis for alcohol-fixed sections, albeit accompanied by non-negligible artifacts.

## Evaluation of errors and biases caused by the use of alcohol-fixed sections

To dissect the errors and biases in DRaqL-SC-3seq from the alcohol-fixed sections, we performed an in-depth comparison of the transcriptome between the freshly dissociated and cell-block cells by taking advantage of the fact that these cells were prepared from the same culture batch of homogeneous 2i-LIF mESCs.

We examined the statistical significance of differences in the detection rate of each gene (i.e., the frequency of cells in which expression of the gene was detected), between these two types of cells. We found that, out of the genes detected in at least one sample (18,992), 5% showed a significant reduction of detection rates in the cell-block cells ($P < 0.01$; detection-rate errors) (Fig S3A and B). Most of the detection-rate errors occurred in genes with relatively low expression levels (95% occurred in log$_2$ [reads per million mapped reads (RPM) +1] <4; i.e., <15 copies/cell).

Next, as mentioned above, we examined the differences of average expression levels between these cells, and found that 0.26% and 3.7% of genes were up- and down-regulated, respectively, in the cell-block cells by >twofold with $P < 0.01$ (expression-level biases) (Fig 2F). These biases were distributed in relatively low expression levels (54% and 97% occurred in log$_2$ [RPM+1] <4 and <6, respectively) (Fig S3C and D). Therefore, DRaqL-SC3-seq showed both detection-rate errors and expression-level biases due to the use of alcohol-fixed sections in only a small fraction of lowly expressed genes overall.

## Exon–exon junction analysis of DRaqL-SC3-seq cDNA amplification

We next asked whether DRaqL-SC3-seq cDNA amplification from the sections was applicable to quantitative expression profiling of exon–exon junctions, by applying the whole cDNAs to sequencing with the Y-shaped adapter (Figs 3A and S4).

The mapping profiles of the Y-shaped adapter sequencing for these cDNAs showed a bias toward the 3′-ends, whereas even the near 5′-end regions showed mapped reads (Fig 3B). To calculate the exon detection rates in expressed genes, we counted the number of detectable exons for each sample (Trimmed Mean of M-values [TMM] >2). In the freshly dissociated cells, we detected an average of 53% of exons (Fig 3C). For protein-coding genes containing ≤20 exons, 55% exons were detected (Fig 3D). It is worth noting that the numbers of exons were ≤20 in ~94% of mouse genes (Fig 3E). Thus, the SC3-seq cDNA amplification method successfully detected about half of the exons in most genes in the freshly dissociated single cells.

Then, we examined the profiles of exon detection rates in the cell-block cells and found a 5% reduction of exon detection rates (48%) compared with the freshly dissociated cells (Fig 3C). In genes with ≤20 exons, more than half of exons were detected (51%) (Fig 3D). This indicates that the Y-shaped adapter sequencing allows exon profiling in alcohol-fixed sections at an efficiency rate comparable with that in freshly dissociated cells.

Next, we counted the reads mapped to the exon–exon junctions and quantified their expression levels (Fig 3F). In total, we detected 25,283 junctions expressed in at least one sample of freshly

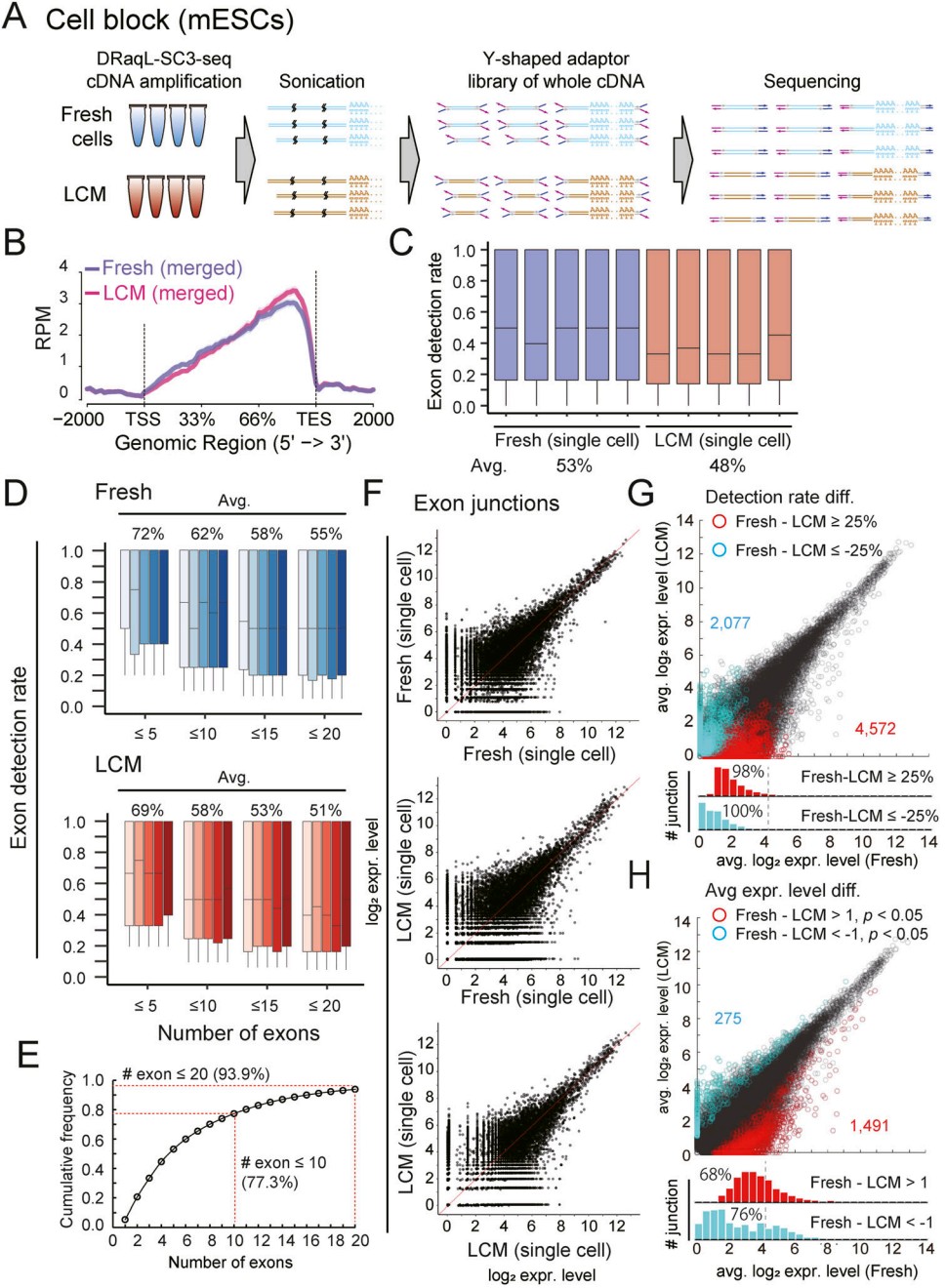

**Figure 3. Exon–exon junction profiling of the DRaqL-SC3-seq cDNA amplification.**
**(A)** Schematic representation of the Y-shaped adapter sequencing of the amplified cDNAs. **(B)** Numbers of mapped reads plotted between the transcription start site and transcription end site. The averages for freshly dissociated single cells (fresh) and single cells from alcohol-fixed cell-block sections (LCM) are shown (n = 5 each). **(C)** Box plots of detection rates of exons in single cells (n = 5 each). **(D)** Box plots of exon detection rates of genes with different numbers of exons (indicated number of exons or fewer) in single cells (n = 5 each). **(E)** Frequency of mouse protein-coding genes that have the indicated numbers of exons or fewer. **(F)** Representative scatterplots of the expression levels of exon–exon junctions between freshly dissociated single cells (fresh) and single cells from alcohol-fixed cell-block sections (LCM). **(G, H)** Scatterplots of averaged exon–exon junction expression levels of freshly dissociated single cells (Fresh) and single cells from alcohol-fixed cell-block sections (LCM) (n = 5 each). **(G, H)** Exon–exon junctions for which the detection-rate difference was ≥25% (G) and the $log_2$ expression difference >1 ($P < 0.05$) (H) are indicated with red and cyan. Histograms of these exon–exon junctions are shown below the scatterplots.
Source data are available for this figure.

dissociated and cell-block cells. Scatterplots showed that the junctional expression levels in single cells were similar between these types of cells. Next, we examined differences in the detection rates and expression levels between freshly dissociated and cell-block cells. We found that most of the >25% difference in the detection rates of junctions occurred at a $log_2$ expression level <4 (Fig 3G). Similarly, differences in expression levels (>twofold, $P < 0.05$) were mainly observed in junctions with a $log_2$ expression level <4 (Fig 3H). Thus, these errors and biases in the exon–exon junction profiling showed trends similar to those in the 3′-sequencing.

Collectively, these results indicate that the DRaqL-SC3-seq cDNA amplification is compatible with a high-quality single-cell transcriptome analysis and exon–exon junction profiling from the alcohol-fixed sections, albeit with bias and errors in the low range of expression levels.

## Application of DRaqL-SC3-seq to mouse ovarian sections

Next, we examined whether DRaqL-SC3-seq can address biological questions, by applying it to alcohol-fixed sections of mouse ovaries (Fig 4A). We analyzed the transcriptome of 44 single growing oocytes

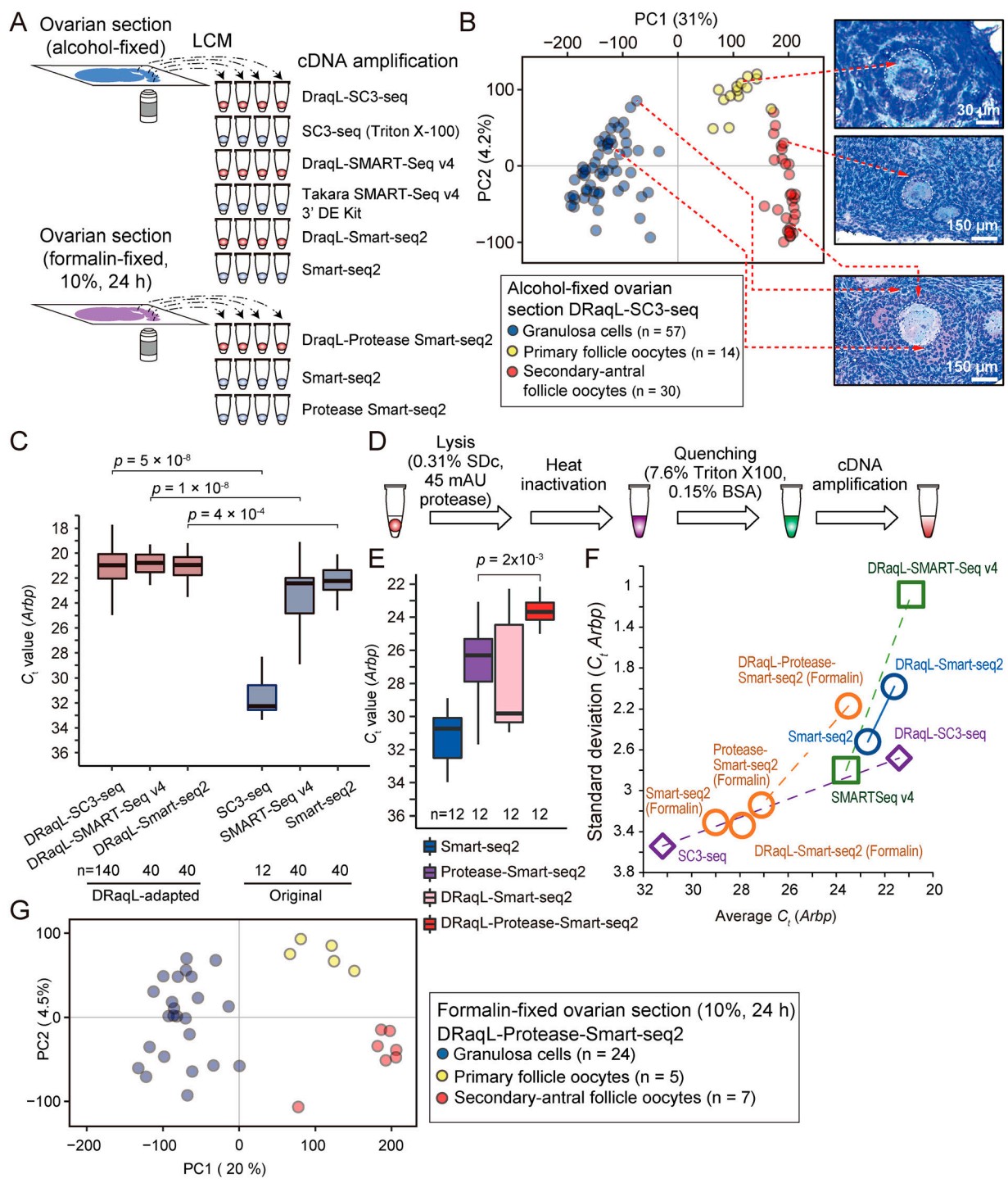

**Figure 4. Application of DRaqL-adapted cDNA amplification methods to mouse ovarian sections and their expansion to formalin-fixed sections.**
**(A)** Schematic representation of different cDNA amplification methods for single cells isolated from alcohol- and formalin-fixed ovarian sections. **(B)** PCA of single oocytes and granulosa cells isolated from mouse ovarian sections analyzed with DRaqL-SC3-seq. Representative histological images are shown to the right of the plots, and isolated cells are indicated with dashed lines. Note that granulosa cells that exhibited a mixed transcriptome profile with oocytes were excluded from subsequent analyses. **(C)** Real-time PCR analysis of cDNAs amplified from single granulosa cells isolated from alcohol-fixed ovarian sections. $Ct$ values of $Arbp$ in cDNAs amplified with the indicated methods are represented with boxplots. $P$-values by a $t$ test are also shown. **(D)** Schematic representation of DRaqL combined with protease treatment. **(E)** Real-time PCR analysis of cDNAs amplified with different Smart-seq2-based cDNA amplification methods from single granulosa cells isolated from formalin-fixed ovarian sections. The $Arbp$ $C_t$ values are represented with boxplots. $P$-values by a $t$ test are shown above the graph. **(F)** Scatterplots of the averages and SDs of the $Arbp$ $C_t$ values in cDNAs amplified with the indicated methods from alcohol- and formalin-fixed sections. The DRaqL-adapted methods are linked with their original methods with dashed lines. The average and SD for DRaqL-Protease-Smart-seq2 and Smart-seq2 from

from primary-to-early antral follicles and 60 single granulosa cells in secondary-to-early antral follicles, isolated with LCM from the sections (Figs S5A–C and S6A and B).

The numbers of detectable genes in single oocytes and granulosa cells were comparable with those in a previous study of the single-cell transcriptome of freshly dissociated human oocytes and granulosa cells, respectively, analyzed with Smart-seq2 (Fan et al, 2021). This suggests that DRaqL-SC3-seq showed sufficient sensitivity for single cells in tissue sections (Fig S5B and D).

In this analysis, we identified a subset of granulosa cells (5%, 3/60) that exhibited expression profiles mixed with those of oocytes in a transcriptome-wide manner (also see the Materials and Methods section; Fig S7A and B). We further validated the presence of mixed expression profiles in freshly dissociated granulosa cells, which were considered to be genuine single cells (see details below; Fig S8). In this study, however, to ensure a detailed investigation of the pure transcriptome of granulosa cells, we excluded these mixed profiles from subsequent analyses, focusing on detailed exploration of the remaining 57 granulosa cells.

As shown in Fig 4B, PCA showed that these oocytes and granulosa cells formed clearly distinct clusters, wherein PC1 represented the difference between these cell types. In addition, scatterplots between serial sections from the same oocytes demonstrated the reproducibility of the analyses (Fig S9A and B).

We also performed RNA-seq of pooled granulosa cells isolated from the sections (10, 100–300, and >300 cells) using the DRaqL-SC3-seq protocol with reduced numbers of PCR cycles and found that the number of detectable genes increased according to the number of pooled cells (Fig S10A and B). Thus, we concluded that DRaqL-SC3-seq is applicable to quantitative transcriptome analysis for single cells and arbitrary sizes of ROIs isolated from alcohol-fixed, mouse ovarian sections.

## Application of DRaqL to other downstream cDNA amplifications

Next, for single granulosa cells isolated from the ovarian sections, we asked whether other cDNA amplification methods can be combined with DRaqL (Fig 4A). First, we examined the use of DRaqL in conjunction with a SMART-Seq v4 3'DE Kit (Takara Bio), a commercially available Smart-seq2-based kit that allows multiplex cDNA library preparation by including index sequences in the reverse transcription primers. We found that its cDNA amplification efficiency was reduced with the use of DRaqL (Fig S11A). In addition, the efficiency was slightly better when the SDc concentration was 0.31% than when it was 0.63%. Thus, using the cell lysis buffer containing 0.31% SDc, we examined the use of additional reverse transcriptases with this kit, and found that the optimal combination (SuperScript II and SuperScript III) was capable of cDNA amplification with efficiency and stability similar to those of the original protocol (Fig S11B). The cDNA amplification efficiency for single granulosa cells isolated from the alcohol-fixed ovarian sections

was similar to that of DRaqL-SC3seq, and we termed this method DRaqL-SMART-Seq v4 cDNA amplification (Fig 4C).

Next, using the combination of separately available reagents and oligo nucleotides with multiplex index sequences, we developed another Smart-seq2-based cDNA amplification method adapted to DRaqL (DRaqL-Smart-seq2; see the Materials and Methods section), and found that this combination achieved cDNA amplification efficiency similar to that by DRaqL-SC3-seq and DRaqL-SMART-Seq v4 (Fig 4C). In fact, for single granulosa cells in the alcohol-fixed sections, DRaqL-SMART-seq v4 and DRaqL-Smart-seq2 improved both the efficiency and stability of cDNA amplification over SMART-Seq v4 3'DE Kit, Smart-seq2, and other LCM-combined cDNA amplification methods (Nichterwitz et al, 2016; Chen et al, 2017; Foley et al, 2019) (Figs 4C and F and S11C and D). Thus, we established DRaqL-adapted, efficient single-cell Smart-seq2-based methods for alcohol-fixed tissue sections.

## Application of DRaqL to formalin-fixed sections

Next, we examined whether DRaqL-adapted cDNA amplification is compatible with formalin-fixed sections. To test the versatility of the method, we used a strong fixative condition, 10% formalin for 24 h at room temperature, which is frequently used in histopathology (Fig 4A and D). We applied DRaqL-Smart-seq2 and Smart-seq2 to mouse ovarian sections fixed with this condition, and found that cDNA amplification efficiency was severely reduced (Fig 4E).

To digest the formalin-fixed cellular components, we combined DRaqL-Smart-seq2 with a thermolabile protease with an optimization of heat inactivation (DRaqL-Protease-Smart-seq2) (Figs 4E and S11E). Application of DRaqL-Protease-Smart-seq2 to single granulosa cells from the formalin-fixed sections resulted in robustly improved cDNA amplification efficiency over those of Smart-seq2, DRaqL-Smart-seq2, and another previous method exploiting proteinase K and additional RNA hydrolysis for cell lysis (Foley et al, 2019) (Figs 4E and F and S11F).

To evaluate the effect of DRaqL on the protease treatment more directly, we compared the cDNA amplification efficiency of DRaqL-Protease-Smart-seq2 with a simple combination of protease and Smart-seq2 (Protease-Smart-seq2), an approach employed in a previous study (Perez et al, 2021). Whereas the amplification efficiency of Protease-Smart-seq2 was better than those of Smart-seq2 and DRaqL-Smart-seq2, DRaqL-Protease-Smart-seq2 showed much better cDNA amplification efficiency and stability than Protease-Smart-seq2, demonstrating that DRaqL significantly improved the efficiency of protease treatment (Fig 4E and F).

Importantly, DRaqL-Protease-Smart-seq2 showed reproducible cDNA amplification efficiency from formalin-fixed sections, with a SD comparable with that of DRaqL-Smart-seq2 from alcohol-fixed sections (Fig 4F). RNA-seq of single oocytes and granulosa cells from the formalin-fixed sections with DRaqL-Protease-Smart-seq2 gave gene expression profiles similar to those from alcohol-fixed sections with DRaqL-SC3-seq (Figs 4G and S11G–I). These data

---

formalin-fixed sections were calculated for the three independent experiments shown in Fig S11F. **(G)** PCA of single oocytes and granulosa cells isolated from the formalin-fixed mouse ovarian sections. **(B)** The color coding is the same as in (B).
Source data are available for this figure.

demonstrate that DRaqL-Protease-Smart-seq2 is capable of robust, high quality transcriptome analysis for single cells in sections strongly fixed with formalin.

It is also worth noting that all cDNA amplification methods developed and examined in this study, as shown in Fig 4 (SC3-seq, SMART-seq v4, Smart-seq2, DRaqL-SC3-seq, DRaqL-SMART-seq v4, DRaqL-Smart-seq2, DRaqL-Protease-Smart-seq2, Protease-Smart-seq2), were performed on sections from the same ovary, ensuring accurate evaluation of the method performance.

### Direct transcriptomic comparison between freshly dissociated oocytes and granulosa cells and those isolated with LCM from ovarian sections

To thoroughly assess the performance of our methods for tissue sections, we isolated single oocytes and granulosa cells with LCM from alcohol-fixed, ovarian-frozen sections obtained from one ovary of a proestrus mouse, and prepared cDNAs using DRaqL-Smart-seq2. Simultaneously, we isolated freshly dissociated oocytes and granulosa cells from the other ovary of the same mouse and prepared cDNAs using Smart-seq2 (Fig S8A).

For the preparation of freshly dissociated cells, we meticulously collected cumulus–oocyte complexes (COCs) from the ovary, and isolated oocytes and their attached single granulosa cells, and other granulosa cells in COCs, through pipetting with a glass capillary under stereomicroscopic inspection. In addition, to prevent the spontaneous activation of oocytes, all processes were performed in the presence of a cAMP analog (Fig S8A).

The number of detectable genes was highly similar between freshly dissociated and LCM-isolated oocytes and granulosa cells, or even better in the LCM-isolated cells (Fig S8B), and was also comparable with the numbers obtained from the LCM-isolated cells with DRaqL-SC-3seq (Fig S5B) and freshly dissociated cells in a previous study (Fan et al, 2021) (Fig S5D). Moreover, PCA showed that the overall transcriptomes of freshly dissociated and LCM-isolated oocytes and granulosa cells were clustered closely together according to cell types (Fig S8C–E). The difference in transcriptome profiles between these cells would be attributable to the differences in mechanical and signaling stimuli encountered during sample preparation processes, and the differences in cell preparation and cDNA amplification methods. These data reinforce the notion that our method enabled a highly sensitive, high-quality transcriptome analysis from tissue sections.

Furthermore, as aforementioned, we observed that a subset of freshly dissociated granulosa cells displayed expression profiles mixed with those of oocytes and exhibited transcriptomes highly distinct from those of other granulosa cells (Fig S8F–I). The mixed profiles were more frequently found in granulosa cells attached to oocytes compared with those not attached to oocytes (43% [6/14] versus 5% [1/21], respectively). These data suggest the presence of genuine mixed expression profiles of granulosa cells and oocytes.

### Morphology-associated transcriptome dynamics of oocytes in follicles

Next, we asked how the morphology of oocytes is associated with their transcriptome, using DRaqL-SC3-seq. Along the PC2 axis of the

transcriptome, the growing oocytes showed expression profiles tightly linked with the follicular morphology, forming clearly different clusters between primary follicles and secondary-to-antral follicles (Fig 4B). Genes previously known to be involved in oogenesis (such as *Obox*, *Oog*, *Oosp1*, *Bmp15*, *Gdf9*, *Izumo1r*, *H1foo*, and *Bcl2l10*) contributed highly to PC2, suggesting that it represented the growth axis of oocytes (Table S1). More importantly, the PC2 values were highly correlated with the sizes of oocytes and follicles (rank correlation coefficient −0.83 for both), indicating a quantitative association between the histological parameters and transcriptome (Figs 5A and S12 and Table S2).

Next, therefore, we investigated the correlations between the expression levels of individual genes and the oocyte diameters (Fig 5B). The expression levels of the genes highly positively and negatively correlated with the diameters ($r > 0.85$ and $r < -0.80$, respectively) were continuously distributed, and sigmoid curves fitting their distribution showed inflection points at similar diameters (44 μm and 46 μm, respectively) (Fig 5C and Table S3). This suggests that oocytes change their transcriptome most dramatically when their sizes have grown to around these values, accompanied by the reduction of *Figla* and *Sohlh1*, genes downregulated during the primordial-to-primary follicle transition (Pan et al, 2005; Hamazaki et al, 2021).

Then, we evaluated the relationship between the transcriptome of oocytes and their diameter by constructing a statistical model (Figs 5D and E and S12A). Using simple regression analyses with PC1 and PC2 as objective variables and diameter as an explanatory variable, we reconstructed the transcriptome of oocytes for every 10 μm of diameter, and examined which reconstructed transcriptome data were best matched with individual oocytes (Fig 5D). We found that the reconstructed transcriptome was best matched with the transcriptome of oocytes with the most similar diameters (Fig 5D and E), suggesting that PC1 and PC2 had sufficient information to link gene expression and the size of oocytes. On the other hand, in 4 out of the 44 oocytes examined, we found that this statistical model yielded data widely discrepant from the observed data; the diameters of the best-matched reconstructed transcriptome were significantly smaller than the diameters of the observed oocytes (>20 μm decrease; Fig 5D). In line with these observations, these oocytes showed expression signatures similar to those of smaller oocytes (Fig 5E).

### Correlation of size–transcriptome relationship and transcriptomic growth of oocytes in follicles

Furthermore, to investigate the association between the size–transcriptome relationship and molecular signatures related to the oocyte growth, we incorporated an RNA-seq dataset for nongrowing, growing, and germinal vesicle (GV) oocytes in mice reported in a previous study (GSE86297) (Gahurova et al, 2017).

We identified 1,286 genes exhibiting dynamic expression profiles during the oocyte growth (Fig S13A and B). Among these genes, 559 were highly expressed in nongrowing oocytes and consistently decreased during oocyte growth. This subset included essential transcription factors for oocyte differentiation (*Figla*, *Sohlh1*, *Sohlh2*) and meiotic genes (*Sycp1*, *Sycp3*, *Smc1b*, *Syce1*). In addition, 727 genes displayed a consistent increase in the growing

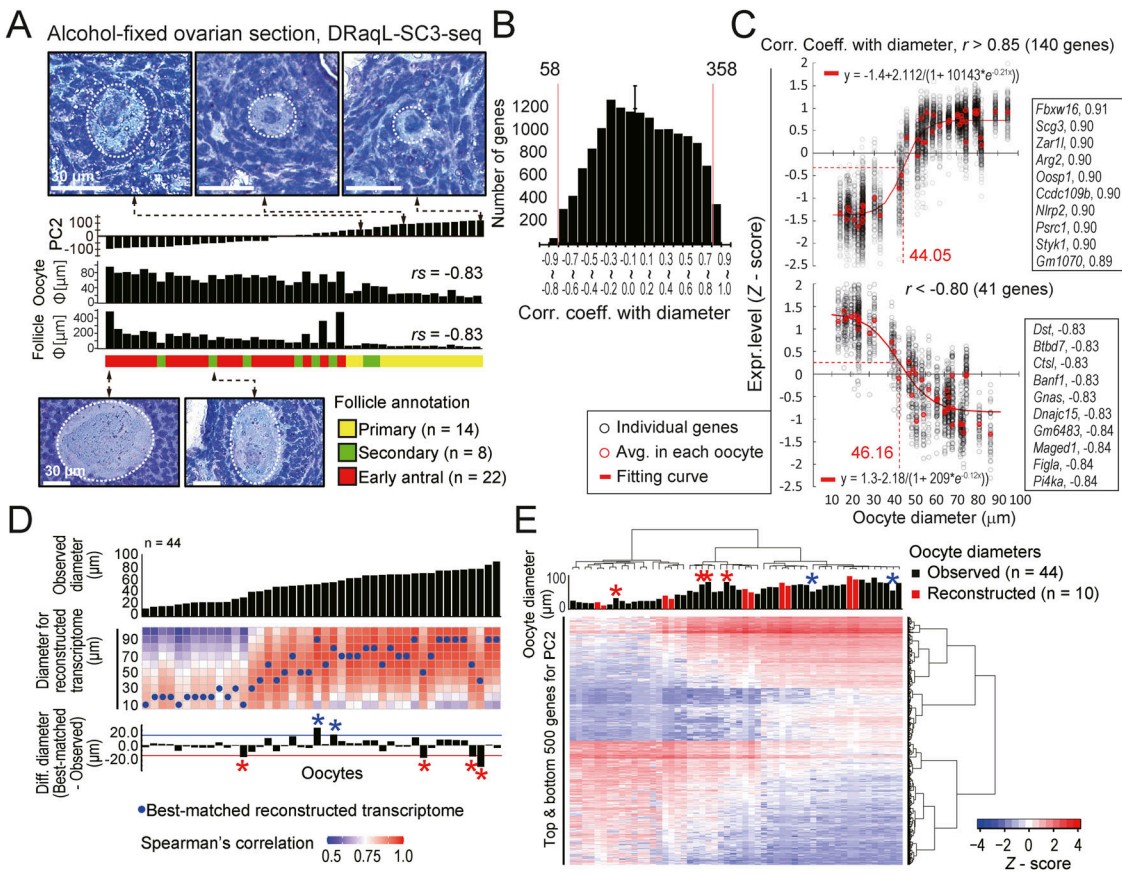

**Figure 5. Morphology-associated transcriptome difference of oocytes revealed with DRaqL-SC3-seq.**
**(A)** Relationship of PC2 values of oocytes isolated from the alcohol-fixed sections and their morphology. PC2 values (top), oocyte diameters (middle), and follicle diameters (bottom) are shown in the bar graphs. The annotations of follicle stages are color coded. Representative histological images of follicles are also shown. Spearman's rank correlation efficient ($rs$) between PC2 and the size of oocytes and follicles is indicated. **(B)** Histogram of Pearson's correlation coefficient between the expression levels of individual genes and oocyte diameters. **(C)** Scatterplots of $\log_2$ gene expression levels against oocyte diameters. Positively correlated genes (top, $r >$ 0.85, n = 140) and negatively correlated genes (bottom, $r <$ −0.80, n = 41) are shown. Average $Z$ scores of these genes in individual oocytes and their fitting curves are plotted with red open circles and red lines, respectively. Gene symbols and $r$ values of the top 10 genes are also indicated. **(D)** Heatmap representation of correlation coefficients between the transcriptome of individual oocytes and reconstructed transcriptome model for the indicated diameters. Diameters of oocytes are shown with the bar graphs. Blue dots indicate the reconstructed transcriptomes best-matched with individual oocytes (i.e., the highest correlation). The difference of diameters between the individual oocytes and the best-matched models are shown with the bar graphs (differences >20 $\mu m$ are indicated with red and blue asterisks). **(E)** Heatmap representation of expression levels of the top 500 genes with positive and negative PC2 values. Diameters of oocytes are shown with the bar graphs, and those of the reconstructed transcriptome model are indicated with red shading. **(D)** Asterisks indicate the same oocytes as in (D).
Source data are available for this figure.

oocytes, reaching their maximum expression levels in GV oocytes, and encompassed oocyte-specific transcription factors (*Obox1*, *Obox2*), a crucial signaling molecule for oogenesis (*Bmp5*), members of the Oogenesin family (*Oog1, Oog2, Oog3, Oog4*), and the DNA methyltransferase essential for the generation of the oocyte epigenome (*Dnmt3l*).

We calculated correlation coefficient of these genes between the previous dataset and our own (Fig S13C). We found that the primary follicle oocytes in our study showed the highest degree of similarity to non-growing oocytes in the previous study, whereas oocytes from secondary-to-early antral follicles showed similarity to growing and GV oocytes in accordance to their respective diameter. Remarkably, the oocytes best matched to the reconstructed transcriptome for a smaller diameter than their actual size, with a decrease of >20 $\mu m$ (Fig 5D), displaying expression profiles similar to those of nongrowing oocytes (Fig S13C). This suggests that these oocytes experienced growth retardation regarding to their transcriptome despite their larger size.

These results indicate that our model successfully reconstructed the transcriptome in the development of normal oocytes, and the deviations from the model delineate heterogeneity of their size–transcriptome relationship relevant to the retardation of transcriptomic maturation, likely reflecting the underlying selective processes for dominant follicles (Deane, 1952; Byskov, 1974).

## Detection of splicing isoforms in single oocytes in sections

Next, we conducted an exon–exon junction analysis of the oocytes in the alcohol-fixed sections, because there is growing evidence of the importance of alternative splicing in oocytes for meiotic

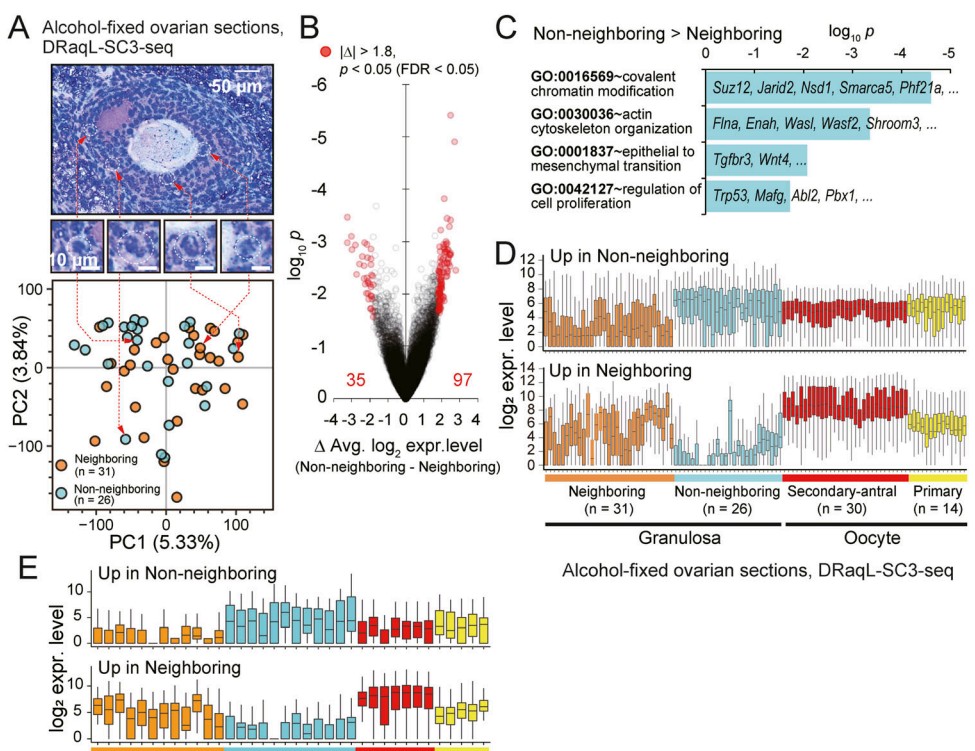

**Figure 6. Single-cell transcriptome analysis of granulosa cells in ovarian sections with DRaqL-SC3-seq.**
**(A)** PCA of DRaqL-SC3-seq transcriptome data of granulosa cells in alcohol-fixed sections. Cells neighboring oocytes (neighboring) and those not neighboring oocytes (non-neighboring) are indicated with orange and cyan circles, respectively. **(B)** Volcano plots of $\log_2$ expression level differences and $\log_{10}$ $P$-values between neighboring and non-neighboring granulosa cells. Genes with differences in $\log_2$ expression levels of more than 1.8 (i.e., >3.5-fold) and $P < 0.05$ with FDR <0.05 are indicated with red circles. **(C)** Bar graphs showing enrichment of gene ontology (GO) terms in genes up-regulated in non-neighboring granulosa cells. **(D)** Box plots of the $\log_2$ expression levels of genes up-regulated in the neighboring and non-neighboring granulosa cells. Expression levels in single oocytes and granulosa cells are shown. **(D, E)** Box plots of the $\log_2$ expression levels of the genes shown in (D) in single oocytes and granulosa cells isolated from the formalin-fixed sections (analyzed with DRaqL-Protease-Smart-seq2). Source data are available for this figure.

progression, oocyte growth and maturation, and female fertility (Tang et al, 2009; Do et al, 2018; Kasowitz et al, 2018; Cheng et al, 2020; Li et al, 2020; Yu et al, 2021) (Fig S14A–H). We applied the Y-shaped adapter sequencing to the single-oocyte cDNAs (n = 5) (Fig S12C), and successfully detected splice isoforms in many genes, including *H1foo*, *Lsm14b*, *Rac1*, *Trp53bp1*, *Oosp1*, and *Parl*. Minor splice isoforms of *Lsm14b*, *Rac1*, and *Trb53bp1*, which are regulated by ESRP1 in oocytes (Yu et al, 2021), were detected in this analysis. Moreover, in *Serf2* and *Cox6b2*, we detected oocyte-specific minor splice isoforms.

## Histology-associated gene expression differences in granulosa cells

Finally, we investigated relationships between the histology and transcriptome of granulosa cells in early antral follicles. Granulosa cells have direct contacts with oocytes through gap junctions on transzonal projections, depending on their positions relative to oocytes (Simon et al, 1997; Li & Albertini, 2013). Thus, we asked whether or not the gene expression profiles of granulosa cells were different between the cells that histologically neighbored oocytes and those that did not—that is, whether there were differentially expressed genes (DEGs) between neighboring (n = 31) and non-neighboring granulosa cells (n = 26) (Fig 6A). The neighboring granulosa cells formed a layer covering the surface of the oocyte, whereas the non-neighboring granulosa cells formed relatively loosened structures compatible with the formation of follicular cavities (Figs S5A and S12B).

Both types of granulosa cells expressed signature genes related to pre-antral cumulus and mitotic antral granulosa cells, as identified in a previous study for estrus murine ovaries (Morris et al, 2022). These cells were negative or low in markers for mural and atretic granulosa cells and corpus luteum (Fig S15), suggesting that they were cumulus granulosa cells in an actively proliferating phase. The absence of atretic follicles in this analysis may reflect the characteristics of the LCM-based cell isolation approach, where only morphologically normal follicles were selected by microscopic inspection before cell isolation.

As shown in Fig 6A, these two types of cells showed no clear transcriptome differences in PCA. However, a direct comparison of gene expression levels revealed 35 and 97 genes that were up-regulated in neighboring and non-neighboring granulosa cells, respectively (average $\log_2$ difference >1.8 [i.e., >~3.5-fold], FDR<0.05) (Fig 6B and Table S4). We found that genes up-regulated in non-neighboring granulosa cells were enriched with genes coding for chromatin modifiers, including Polycomb Repressive Complexes (*Suz12*, *Jarid2*), regulators of chromatin modifiers (*Ogt*, *Nsd1*), a histone H3-lysine4-mediated regulator (*Ph421a*), and a SWI/SNF family member (*Smarca5*) (Fig 6C). In addition, these cells were also enriched with genes involved in actin cytoskeleton organization (*Flna*, *Enah*), epithelial-to-mesenchymal transition (*Tgfbr3*, *Wnt4*), and regulation of cell proliferation (*Trp53*, *Abl2*, *Pbx1*), suggesting that their histological positions were associated with distinct proliferative and morphological characters, indicative of granulosa cell differentiation.

On the other hand, the 35 genes up-regulated in the neighboring granulosa cells showed highly heterogeneous expression levels even among these cells, whereas they were nearly undetectable in the non-neighboring granulosa cells (Fig 6D). Interestingly, they included oocyte-specific genes, such as *Obox* family members, which were expressed in oocytes in the secondary-to-early antral follicles at much higher expression levels. These results most likely reflected gap junction communications in the oocyte and granulosa cells (Anderson & Albertini, 1976; Simon et al, 1997; Matzuk et al, 2002). In addition, it is worth noting that the expressions of these oocyte-specific genes occurred in a gene-specific manner, but were not due to transcriptome mix with oocytes, which were excluded from this analysis as aforementioned (Figs S7B and S8H).

RNA-seq of 10 pooled granulosa cells from early antral follicles confirmed the expression profiles of these DEGs, supporting the accuracy of the histology-linked gene expression analysis at the single-cell level with DRaqL-SC3-seq (Fig S10A and C). In bulk RNA-seq of whole granulosa from early antral follicles, these DEGs were expressed at intermediate levels, consistent with the idea that bulk granulosa is a mixture of these cell types. These results demonstrate that these DEGs can be identified only by identifying their histological affiliation to oocytes followed by a transcriptome analysis.

In addition, DRaqL-Protease-Smart-seq2 showed that the expression profiles of these DEGs in the single granulosa cells from the formalin-fixed sections were similar to those in the single granulosa cells from the alcohol-fixed sections, again demonstrating the robust performance of this method for the formalin-fixed sections (Fig 6E).

### Expression patterns of DEGs in granulosa cells in previous single-cell RNA-seq datasets

We further investigated the expression patterns of the aforementioned DEGs, which consisted of the up-regulated genes in the neighboring and non-neighboring granulosa cells (Fig 6B), using previously published single-cell transcriptome data for mouse ovarian somatic cells (Li et al, 2021) and human COCs (Fan et al, 2021) (Fig S16). In these previous datasets, we were able to identify two groups of granulosa cells, each representing the top and bottom 25% based on the expression level difference between these DEG groups. Notably, the up-regulated genes in neighboring and non-neighboring cells were also differentially expressed between these top and bottom 25% cell groups with statistical significance ($P < 0.001$ with *Wilcoxon* test) (Fig S16A–D). These results demonstrate that DRaqL-SC3-seq for ovarian sections revealed previously unidentified co-expression patterns in granulosa cells.

In addition, in the dataset of human COCs (Fan et al, 2021), we observed that one of the granulosa cells displayed an expression profile highly similar to the oocyte transcriptome, suggesting the presence of a mixed profile. We also observed several oocyte-specific genes expressed in the human granulosa cells consistent with the previous report (Fan et al, 2021) (Fig S16E and F). These results further support our findings on the oocyte-specific genes expressed in granulosa cells described above.

### Protein expression patterns of DEGs in granulosa cells

To validate the findings of the DRaqL-SC3-seq regarding the differential gene expression in granulosa cells within ovarian sections, we employed an independent approach by examining the protein expression patterns of these DEGs through immunofluorescence. Specifically, we analyzed *Pbx1* and *Suz12*, which were found to be up-regulated in the non-neighboring granulosa cells as mentioned above. Consistent with the DRaqL-SC3-seq data, immunofluorescence analysis revealed significantly higher levels of PBX1 and SUZ12 protein signals in non-neighboring granulosa cells than in neighboring cells in antral follicles ($P = 1.5 \times 10^{-4}$ and $2.4 \times 10^{-3}$, respectively, by *Wilcoxon* test) (Fig S17A–C). These data provide additional support for the histology-associated gene expression patterns identified through DRaqL-SC3-seq.

Collectively, these results demonstrated that single-cell RNA-seq with DRaqL from ovarian sections allowed quantitative analyses of histology-based expression differences that cannot be revealed by transcriptomics alone, and splice isoform analyses.

# Discussion

In this study, we developed robust and quantitative cDNA amplification methods combined with cell lysis by DRaqL for single cells isolated from alcohol- and formalin-fixed sections using LCM. Single cells were isolated from tissue sections under careful microscopic inspection, although the possibility of contamination from adjacent cells within the same section could not be excluded. The results of quantitative evaluation revealed that DRaqL-SC3-seq from alcohol-fixed sections showed only small errors and biases occurring in relatively lowly expressed genes, and showed amplification efficiency similar to those of the DRaqL-adapted Smart-seq2-based methods, suggesting the versatility of DRaqL.

Combining DRaqL with protease treatment also enabled robust cDNA amplification and RNA-seq from sections fixed with formalin, the most popular fixative for tissue preservation (Titford & Horenstein, 2005; Paavilainen et al, 2010) (Fig 4). The strong fixative condition used in this study, 10% formalin for 24 h at room temperature, is preferred in histopathology (Titford & Horenstein, 2005). Both cDNA amplification efficiency and stability in our method were significantly improved compared with the previous approaches, suggesting that DRaqL would facilitate cell lysis by protease.

For efficient cell lysis compatible with subsequent enzymatic reactions, DRaqL takes advantage of the efficient incorporation of a small amount of denaturing detergent into micelles of non-denaturing detergents (Jonstromer & Strey, 1992; Sivars et al, 1994). This strategy has been used for the extraction of bacterial genomic DNA with SDS, followed by region-specific PCR after quenching with Tween20 (Goldenberger et al, 1995). This tactic has also been employed in the Hi-C technique, in which SDS is used to de-condense genomic DNAs followed by quenching with Triton X-100, in preparation for subsequent reactions such as restriction enzyme treatment (Lieberman-Aiden et al, 2009). As another example, a mix of SDS and nonionic detergents has been used for

cellulase-mediated glycolysis (Eriksson et al, 2002). In this study, we applied the combination of a denaturing detergent and quenching to RNA-seq of alcohol- and formalin-fixed sections and rigorously evaluated the performance of the method, expanding the application range of this principle in quantitative biology.

By applying this method to alcohol-fixed sections of mouse ovaries, we successfully depicted histology-associated transcriptional differences of oocytes and granulosa cells (Figs 4, 5, and 6). The formation of discrete clusters of oocytes in primary follicles and secondary-to-early antral follicles may reflect transcriptomic changes during the primary-to-secondary follicle transition (Williams & Erickson, 2000; Zhang et al, 2018). The quantitative continuum of the size–transcriptome relationships of oocytes allowed us to realize the morphology-based prediction of the transcriptome (Fig 5), providing a significantly higher resolution of this relationship than in a previous study in which dissociated oocytes were classified by diameter into three clusters and the DEGs were identified (Gu et al, 2019). It is worth noting that the transcriptome data obtained in this study were associated with the intact morphology of snap-frozen ovarian tissues, without any artifacts caused by cell dissociation. The heterogeneity of the relationship between the transcriptome of oocytes and their size as revealed using our statistical model might help elucidate the mechanisms of dominant follicle selection and quality control of oocytes. Furthermore, for granulosa cells, we identified genes that were differentially expressed depending on their positions within the follicles, which suggests distinct epigenetic regulation and cell-cycle activities (Fig 6). Thus, transcriptome analysis with DRaqL revealed a histology-associated, quantitative difference of transcriptomic profiles at the single-cell level.

A previous study by Ernst and colleagues conducted an LCM-based transcriptomic analysis of human oocytes during the primordial-to-primary follicle transition (Ernst et al, 2017). The oocyte transcriptome was analyzed using RNA that was extracted and purified from 45–186 pooled oocytes, whereas our methods enable quantitative transcriptome analysis of LCM-isolated single oocytes and granulosa cells without the need for RNA purification. In a comparative analysis of gene expression profiles between human and mouse oocytes, more than half of the genes that were differentially expressed during human primordial-to-primary follicle transition had similar expression levels in both species (Fig S18A and B). However, genes related to the mTOR and GnRH signaling pathways exhibited differential expression levels between human and mouse oocytes. In addition, FOXO1, a key transcription factor down-regulated in human primary-follicle oocytes, was expressed at a higher level in mice (Fig S18B). These findings may indicate molecular differences in the growth of human and mouse oocytes.

In previous studies, direct application of the Smart-seq2 cDNA amplification, with non-denaturing cell lysis, has been employed for alcohol-fixed sections (Nichterwitz et al, 2016; Brasko et al, 2018; Deng et al, 2019; Lee et al, 2022). The lysis efficiency, which may depend on various parameters such as fixation conditions, has been controversial, and a purification-based approach has been proposed (Chen et al, 2017). In this study, our DRaqL-adapted methods achieved an improved cDNA amplification over the previous methods using non-denaturing cell lysis or RNA purification (Nichterwitz et al, 2016; Chen et al, 2017; Foley et al, 2019; Perez et al, 2021) (Figs 4 and S11).

The cDNA amplification methods used in this study were conducted in individual tubes, and thus, their throughput would be comparable with those of previous studies using similar cDNA amplification approaches (e.g., the recent single-cell/low-input studies for primate gastrulae with Smart-seq2 [~2,000 cells/samples] [Tyser et al, 2021; Bergmann et al, 2022]). In addition, the throughput could be further improved by using the multiplexed strategy employed in DRaqL-SMART-seq v4, DRaqL-Smart-seq2, and DRaqL-Protease-Smart-seq2.

On the other hand, the throughput might be limited by the one-by-one approaches for the LCM-based cell isolation, which are dependent on target tissues and the ease of identifying cells of interest. In this study, for the granulosa cells, the whole process of identifying target follicles and isolating cells of interest was manually conducted across many sections (typically, 1–2 target follicles per ovarian section met our investigation criteria). On the other hand, a high-throughput, automated versatile LCM-based cell isolation method has been developed (Brasko et al, 2018) (>1,000 cells per day), which might improve the throughput of analysis.

In spite of the relatively low throughput of its cell-isolation process, one of the advantages of LCM-based spatial transcriptomics over high-throughput methods for whole sections (see Liao et al [2021] for review) might be that it allows deeper transcriptomic analyses for any single cells of interest, including those of splice isoforms. A previous study aligned the cDNA sequences of ROIs to exons for investigation of the epi-transcriptome, but the quantitative performance for the expression levels of exon junctions remains elusive, with relatively low sensitivity of the method (a few thousand genes were detectable in slide-mounted culture cells in this previous report) (Lee et al, 2022). In this study, we performed quantitative exon–exon junction profiling of single cells with deep sequencing (Fig 3) and detection of oocyte-specific splice isoforms (Fig S14), after 3'-end analyses with lower sequencing depths, thereby establishing a flexible single-cell experimental design for in situ transcriptomics. In addition, with respect to this particular application, our methods would have a significant advantage over a previous two-way LCM-combined method compatible with alcohol- and formalin-fixed sections (Foley et al, 2019), which relies on RNA hydrolysis and a short PCR-elongation step, and thus restricts the analysis strictly to the 3' ends of mRNA.

In conclusion, we have proposed an efficient, flexible analytical framework for single-cell transcriptomics from alcohol- and formalin-fixed tissue sections, which would serve as a complemental approach to the current high-throughput spatial and single-cell transcriptomics, as demonstrated by the discovery of histology-associated transcriptomic heterogeneity in the growing ovarian follicles.

# Materials and Methods

### Cell culture

mESCs were cultured in a medium containing 2iLIF (GSK inhibitor [CHIR99021; 3 $\mu$M], MEK inhibitor [PD0325901; 0.4 $\mu$M], and LIF [1 U/$\mu$l] [2iLIF]) as described previously (Hayashi et al, 2011).

## Sectioning of frozen cell blocks

To make frozen cell blocks, the mESCs were pelleted and washed with PBS. The mESCs were then suspended in 1 × PBS containing 10% PVA (Sigma-Aldrich), embedded in a cryomold, and snap-frozen in liquid nitrogen ($N_2$). Subsequently, the frozen cell blocks were sectioned at a thickness of 15 $\mu$m, using a CM1860UV cryostat (Leica), and were mounted on MembraneSlide 1.0 PEN slides (Carl Zeiss). The sectioned cell blocks were dried overnight at room temperature and fixed with 80% ethanol. The fixed sections were then stained with 1% cresyl violet acetate (MP Biomedicals) dissolved in 80% ethanol, washed with 100% ethanol, and dried for more than 1 h. For the histological images, contrast and brightness were optimized linearly with Photoshop for the ease of visual inspection. These procedures are summarized in Fig 1A.

## Sectioning, fixation, and staining of mouse ovaries

Mouse ovaries were excised from 8-wk-old female mice and snap-frozen in liquid $N_2$. Then, the frozen ovaries were sunk in 10% PVA in cryomolds and again frozen in liquid $N_2$. The embedded ovaries were sectioned using a CM1860UV cryostat at a thickness of 15 $\mu$m, dried at room temperature for more than 1 h, fixed and stained with 1% cresyl violet acetate dissolved in 50% isopropanol, washed with 100% isopropanol, and dried for more than 1 h. Formalin fixation was performed with 10% formalin neutral buffer solution (Wako) at room temperature for 24 h, and the fixed sections were stained with 1% cresyl violet acetate in 50% isopropanol and dried for more than 1 h.

## Laser capture microdissection

The mESCs on the frozen sections were isolated using a laser microdissection system PALM MB4 (Zeiss) with a ×20 objective lens. The dissection laser setting, cutting speed, and laser pulse catapulting (LPC) were set as 46%, 50%, and 54%, respectively. The dissected cells were collected into the lysis buffer in the caps of single, flat-top 200-$\mu$l PCR tubes (Greiner Bio-One).

In mouse ovarian sections, we analyzed follicle morphology using PALM MB4 microscopy. Specifically, we selected normally growing primary follicles and secondary-to-early antral follicles in sections containing nuclei of oocytes for subsequent transcriptome analysis. To isolate oocytes and granulosa cells, we identified single granulosa cells and oocytes under microscopic inspection with a ×60 objective lens. To avoid collecting granulosa cells overlapped within a single section, we carefully inspected the morphology of the nuclei and drew lines to guide the cutting laser, encircling the single nuclei using the "Joint Cut" mode. Given that the thickness of the sections was 15 $\mu$m, which is similar to the size of a single granulosa cell, we consider it unlikely that two or more entire cells were isolated simultaneously with this procedure. The parameters used for isolating oocytes and granulosa cells were as follows: cutting energy at 44%, LPC energy at 30%, and speed at 40. The dissected cells were collected into the lysis buffer in the caps of single, flat-top 200-$\mu$l PCR tubes. These LCM procedures are summarized in Fig 1A and the results of LCM on ovarian sections are shown in Fig S6A and B.

## DRaqL-SC3-seq cDNA amplification

Cells in alcohol-fixed sections were isolated with LCM in 6.4 $\mu$l of cell lysis buffer (0.8 $\mu$l of GeneAmp 10xPCR Buffer II [Thermo Fisher Scientific], 0.48 $\mu$l of 25 mM $MgCl_2$ [Thermo Fisher Scientific], 0.8 $\mu$l of 5% sodium deoxycholate [SDc] [Nacalai Tesque], 0.4 $\mu$l of 100 mM dithiothreitol [DTT] [Thermo Fisher Scientific], 0.64 $\mu$l of 40 U/$\mu$l RNaseOUT [Thermo Fisher Scientific], 0.08 $\mu$l of 40 U/$\mu$l porcine liver RNase inhibitor [Takara Bio], 0.16 $\mu$l of 2.5 mM dNTP [Takara Bio], 0.16 $\mu$l of 1:500,000 diluted ERCC RNA Spike-In Mix 1 [Thermo Fisher Scientific], 0.16 $\mu$l of 10 ng/$\mu$l V1[dT]$_{24}$ primer [Hokkaido System Science], and 2.72 $\mu$l of deionized distilled water [DDW, Gibco]) in the flat caps of 0.2-ml PCR tubes (Greiner Bio-One), and spun down into the tubes by brief centrifugation. Cells were then lysed at 70°C for 6 min, followed by the addition of 2.8 $\mu$l quenching buffer (0.7 $\mu$l of Triton X-100 [Nacalai Tesque], 0.7 $\mu$l of 2% BSA [Takara Bio]) and incubation at 70°C for 90 s to quench the denaturing effect of SDc.

The cDNA synthesis and amplification were performed as described previously (Kurimoto et al, 2006) with minor modifications. In brief, the first-strand cDNA was synthesized by adding 0.5 $\mu$l of reverse transcriptase mix (0.2 $\mu$l of 200 U/$\mu$l SuperScript III [Thermo Fisher Scientific], 0.033 $\mu$l of 40 U/$\mu$l porcine liver RNase inhibitor, 0.067 $\mu$l of 1–10 mg/ml T4 gene 32 product [Roche], and 0.2 $\mu$l of DDW [Gibco]) to the cell lysate (9.2 $\mu$l) and incubating at 50°C for 5 min followed by heat inactivation at 70°C for 10 min. The excess primers were degraded by adding 1 $\mu$l of exonuclease I mixture (1×exonuclease I buffer and 0.5 U exonuclease I [Takara Bio]) and incubating at 37°C for 30 min followed by heat inactivation at 80°C for 25 min. Then, the first-strand cDNA product was poly dA-tailed by adding 3 $\mu$l TdT mixture (0.6 $\mu$l of 10× PCR Buffer II, 0.36 $\mu$l of 25 mM $MgCl_2$, 0.18 $\mu$l of 100 mM dATP, 0.3 $\mu$l of 15 U/$\mu$l recombinant terminal deoxynucleotidyl transferase [Thermo Fisher Scientific], 0.3 $\mu$l of 2 U/$\mu$l RNaseH [Thermo Fisher Scientific], and 1.26 $\mu$l of DDW) and incubating at 37°C for 1 min followed by heat inactivation at 70°C for 10 min. The dA-tailed product (13.7 $\mu$l) was divided into four tubes, and second-strand cDNA was synthesized by adding 9.5 $\mu$l V3-PCR mixture (0.95 $\mu$l of 10× ExTaq Buffer [Takara Bio], 0.95 $\mu$l of 2.5 mM dNTP, 0.1875 $\mu$l of 1 $\mu$g/$\mu$l V3[dT]$_{24}$ primer, 0.095 $\mu$l of ExTaq Hot Start version [ExTaqHS, Takara Bio], and 7.3175 $\mu$l of DDW) and applying thermal cycling program 1. Then, cDNA amplification was performed by adding 9.5 $\mu$l of V1-PCR mixture (0.95 $\mu$l of 10× ExTaq Buffer [Takara Bio], 0.95 $\mu$l of 2.5 mM dNTP, 0.1875 $\mu$l of 1 $\mu$g/$\mu$l V1[dT]$_{24}$ primer, 0.095 $\mu$l of ExTaqHS, and 7.3175 $\mu$l of DDW) and applying thermal cycling program 2. For oocytes, we used a PCR program with 18 cycles. The amplified cDNA (~89.7 $\mu$l) was purified with a 0.6× volume of AxyPrep MAG PCR clean-up reagent (Axygen) according to the manufacturer's instructions. All oligonucleotides and thermal cycling programs used in this study are listed in Table S5. DRaqL and adapted cDNA amplification are schematically represented in Fig 1A and B.

## DRaqL-SMART-Seq v4 cDNA amplification

A Smart-seq v4 3′ DE Kit (Takara Bio) was adapted to DRaqL as follows: cells in alcohol-fixed sections were isolated with LCM in 6.4 $\mu$l of cell lysis buffer (0.25 $\mu$l of 40 U/$\mu$l Takara RNase Inhibitor, 0.4 $\mu$l of 100 mM DTT, 0.16 $\mu$l of 1:500,000 diluted ERCC RNA Spike-In

Mix 1, 0.4 $\mu$l of 5% SDc, and 5.19 $\mu$l of nuclease-free water) in the caps of PCR tubes, and spun down into the tubes by brief centrifugation. Cells were then lysed at 72°C for 3 min, followed by the addition of 2.8 $\mu$l quenching buffer (0.7 $\mu$l of Triton X-100, 0.7 $\mu$l of 2% BSA, and 1.4 $\mu$l of 5× SuperScript II buffer [Thermo Fisher Scientific]) to reduce the denaturing effect of SDc. Then, 2.8 $\mu$l of RT oligo (1 $\mu$l of 12 $\mu$M Oligo dT In-line Primer and 1.8 $\mu$l of nuclease-free water: kit components) was added, followed by incubation at 72°C for 90 s. Reverse transcription was performed by adding 7.5 $\mu$l of Master Mix (4 $\mu$l of 5× ultra-low first-strand buffer, 1 $\mu$l of 48 $\mu$M SMART-Seq V4 oligonucleotide, 0.5 $\mu$l of 40 U/$\mu$l RNase inhibitor, and 2 $\mu$l of SMARTScribe Reverse Transcriptase: kit component) supplemented with 0.25 $\mu$l of SuperScript II and 0.25 $\mu$l of SuperScript III, and by incubating at 42°C for 90 min and 70°C for 10 min. Then, cDNA amplification was performed by adding 30 $\mu$l of kit component PCR Mix (25 $\mu$l of 2× SeqAmp PCR Buffer, 1 $\mu$l of 12 $\mu$M Blocked PCR Primer II A, 1 $\mu$l of SeqAmp DNA polymerase, and 3 $\mu$l of nuclease-free water) and applying the thermal cycling program for SMART-seq v4. The amplified cDNA (50 $\mu$l) was purified with a 0.8× volume of AxyPrep MAG PCR clean-up reagent according to the manufacturer's instructions.

### DRaqL-smart-seq2 cDNA amplification

Cells in alcohol-fixed sections were isolated with LCM in 6.4 $\mu$l of cell lysis buffer (0.6 $\mu$l of 5× SuperScript II buffer, 0.1 $\mu$l of 100 $\mu$M Oligo dT VN, 0.8 $\mu$l of dNTP mix [25 mM each], 0.25 $\mu$l of 40 U/$\mu$l recombinant RNase inhibitor [Takara Bio], 0.5 $\mu$l of 100 mM DTT, 0.06 $\mu$l of 1 M MgCl$_2$ [Sigma-Aldrich], 2 $\mu$l of 5 M Betaine [Sigma-Aldrich], 0.16 $\mu$l of 1:500,000 diluted ERCC RNA Spike-In Mix 1, 0.4 $\mu$l of 5% SDc, and 1.53 $\mu$l of DDW) in the caps of PCR tubes, and spun down into the tubes by brief centrifugation. The isolated cells were lysed at 72°C for 6 min, followed by addition of 2.8 $\mu$l quenching buffer (0.7 $\mu$l of Triton X-100, 0.7 $\mu$l of 2% BSA, and 1.4 $\mu$l of 5× SuperScript II buffer [Thermo Fisher Scientific]). Then, 1.6 $\mu$l of a template-switching mixture (0.1 $\mu$l of 100 $\mu$M N-template-switching oligo, 0.25 $\mu$l of SuperScript II, 0.25 $\mu$l of SuperScript III, 0.2 $\mu$l of recombinant RNase inhibitor, and 0.8 $\mu$l of DDW) was added, followed by the cycling RT program for DRaqL-Smart-seq2. Then, cDNA amplification was performed by adding 15 $\mu$l of Seq amp PCR mixture (12.5 $\mu$l of 2x SeqAmp buffer [Takara Bio], 0.05 $\mu$l of 100 $\mu$M N-IS PCR primer, 0.5 $\mu$l of SeqAmp DNA polymerase [Takara Bio], and 1.95 $\mu$l of DDW) and applying the thermal cycling program for SeqAmp. For oocytes, we used a PCR program of 18 cycles. The amplified cDNA (25.8 $\mu$l) was purified with a 0.8× volume of AxyPrep MAG PCR clean-up reagent according to the manufacturer's instructions.

### DRaqL-Protease-smart-seq2 cDNA amplification

Cells in formalin-fixed sections were isolated with LCM in 6.4 $\mu$l of cell lysis buffer (0.6 $\mu$l of 5× SuperScript II buffer, 0.1 $\mu$l of 100 $\mu$M Oligo dT VN, 0.8 $\mu$l of dNTP mix (25 mM each), 0.25 $\mu$l of 40 U/$\mu$l recombinant RNase inhibitor, 0.5 $\mu$l of 100 mM DTT, 0.06 $\mu$l of 1 M MgCl$_2$, 2 $\mu$l of 5 M Betaine, 0.4 $\mu$l of 5% SDc, 0.32 $\mu$l of 900 mAU/ml QIAGEN Protease, and 1.37 $\mu$l of DDW) in the caps of PCR tubes, and spun down into the tubes by brief centrifugation. The isolated cells were lysed by protease digestion at 50°C for 10 min followed by heat

inactivation at 80°C for 15 min. The denaturing effect of SDc was quenched by addition of 0.8 $\mu$l of 1:2,500,000 diluted ERCC RNA Spike-In Mix 1 and 2.8 $\mu$l of quenching buffer (0.7 $\mu$l of Triton X-100, 0.7 $\mu$l of 2% BSA, and 1.4 $\mu$l of 5× SuperScript II buffer [Thermo Fisher Scientific]), followed by incubation at 72°C for 90 s. Then, 0.8 $\mu$l of a template-switching mixture (0.1 $\mu$l of 100 $\mu$M N-template-switching oligo, 0.25 $\mu$l of SuperScript II, 0.25 $\mu$l of SuperScript III, and 0.2 $\mu$l of recombinant RNase inhibitor) was added, followed by the cycling RT program for DRaqL-Smart-seq2. Then, cDNA amplification was performed by adding 15 $\mu$l of the SeqAmp PCR mixture, and applying the thermal cycling program for SeqAmp. The amplified cDNA (25.8 $\mu$l) was purified with a 0.8× volume of AxyPrep MAG PCR clean-up reagent according to the manufacturer's instructions.

### SC3-seq cDNA amplification with Triton X-100

To evaluate cell lysis efficiency with DRaqL, the cell lysis step of DRaqL-SC3-seq was replaced with cell lysis with Triton X-100 only (SC3-seq cDNA amplification [Triton X-100]) as follows. Cells were isolated with LCM in 6.4 $\mu$l of cell lysis buffer (0.8 $\mu$l of GeneAmp 10xPCR Buffer II, 0.48 $\mu$l of 25 mM MgCl$_2$, 0.8 $\mu$l of 5% Triton X-100, 0.4 $\mu$l of 100 mM DTT, 0.64 $\mu$l of 40 U/$\mu$l RNaseOUT, 0.08 $\mu$l of 40 U/$\mu$l porcine liver RNase inhibitor, 0.16 $\mu$l of 2.5 mM dNTP, 0.16 $\mu$l 1:500,000 diluted ERCC RNA Spike-In Mix 1, 0.16 $\mu$l of 10 ng/$\mu$l V1[dT]$_{24}$ primer, 2.72 $\mu$l of DDW) in the flat caps of PCR tubes, and spun down into the tubes by brief centrifugation. Cells were then lysed at 70°C for 6 min, followed by the addition of 2.8 $\mu$l quenching buffer (0.7 $\mu$l of Triton X-100, 0.7 $\mu$l of 2% BSA) and incubation at 70°C for 90 s to quench the denaturing effect of SDc. The following steps were performed as in DRaqL-SC3-seq.

### SC3-seq cDNA amplification with NP-40

As a positive control of the cDNA amplification of 10-pg total RNA purified from 2i-LIF mESCs, we performed the original protocol of the SC3-seq cDNA amplification with the same procedure as described in Kurimoto et al (2006), with minor modifications. Briefly, 0.5 $\mu$l of 25 pg/$\mu$l total RNA were added to 4 $\mu$l of cell lysis buffer (0.8 $\mu$l of GeneAmp 10xPCR Buffer II, 0.48 $\mu$l of 25 mM MgCl$_2$, 0.8 $\mu$l of 5% NP-40, 0.4 $\mu$l of 100 mM DTT, 0.64 $\mu$l of 40 U/$\mu$l RNaseOUT, 0.08 $\mu$l of 40 U/$\mu$l Porcine Liver RNase Inhibitor, 0.16 $\mu$l of 2.5 mM dNTP, 0.16 $\mu$l 1:500,000 diluted ERCC RNA Spike-In Mix 1, 0.16 $\mu$l of 10 ng/$\mu$l V1[dT]$_{24}$ primer, 2.72 $\mu$l of DDW). The mixture was incubated at 70°C for 6 min, followed by the addition of 2.8 $\mu$l DDW, and then was subjected to first-strand cDNA synthesis. The following steps were performed as in DRaqL-SC3-seq.

### SMART-Seq v4 3′DE Kit cDNA amplification

The cDNA amplification using SMART-Seq v4 3′DE Kit was performed according the manufacturer's instruction (Takara Bio), except for the step of cell capture. 20 $\mu$l of 10× reaction buffer was prepared (19 $\mu$l of 10×Lysis Buffer, 1 $\mu$l of RNase Inhibitor), of which 1 $\mu$l was mixed with 10.5 $\mu$l DDW, resulting in 11.5 $\mu$l of cell lysis buffer. Then, 6.4 $\mu$l of cell lysis buffer was added the flat caps of PCR tube, and cells were isolated with LCM into the reaction buffer, followed by the brief centrifugation. Then, another 5.1 $\mu$l of cell lysis buffer was

added to the tubes (11.5 $\mu$l in total). The following steps were performed exactly according to the manufacturer's instruction. Briefly, 1 $\mu$l of 12 $\mu$M Oligo dT In-line Primer was added to the mixture and incubated at 72°C for 3 min. Then, reverse transcription was performed by adding 7.5 $\mu$l of Master Mix (4 $\mu$l of 5× ultra-low first-strand buffer, 1 $\mu$l of 48 $\mu$M SMART-Seq V4 Oligonucleotide, 0.5 $\mu$l of 40 U/$\mu$l RNase Inhibitor, 2 $\mu$l of SMARTScribe Reverse Transcriptase) and by incubating at 42°C for 90 min and 70°C for 10 min. Then, cDNA amplification was performed by adding 30 $\mu$l of kit component PCR Mix (25 $\mu$l of 2× SeqAmp PCR Buffer, 1 $\mu$l of 12 $\mu$M Blocked PCR Primer II A, 1 $\mu$l of SeqAmp DNA Polymerase, and 3 $\mu$l of nuclease-free water) and applying the thermal cycling program for SMART-seq v4.

### Smart-seq2 cDNA amplification

Smart-seq2 cDNA amplification was performed as described (Nichterwitz et al, 2016), except for the volume of cell lysis solution. Briefly, cells were isolated with LCM in 6.4 $\mu$l cell lysis solution (1 $\mu$l of 10 $\mu$M Oligo dT VN, 1 $\mu$l of dNTPs mix [10 mM each], 0.21 $\mu$l of 40 U/$\mu$l Recombinant RNase inhibitor, 0.17 $\mu$l of 5% Triton X-100, 0.16 $\mu$l of 1:500,000 diluted ERCC RNA Spike-In Mix 1, and 3.86 $\mu$l of DDW) in the caps of PCR tubes, and spun down into the tubes by brief centrifugation. The isolated cells were lysed at 72°C for 3 min, and were put on ice immediately after the cell lysis. Then, 5.45 $\mu$l of the reverse transcription mixture (2 $\mu$l of 5× SuperScript II buffer, 0.5 $\mu$l of 100 mM DTT, 2 $\mu$l of 5 M Betaine, 0.1 $\mu$l of 1 M MgCl$_2$, 0.25 $\mu$l of 40 U/$\mu$l Recombinant RNase inhibitor, 0.1 $\mu$l of 100 $\mu$M template-switching oligo, and 0.5 $\mu$l of SuperScript II) was added to the tube, followed by cycling RT program for Smart-seq2. Then, cDNA amplification was performed by adding 15 $\mu$l of KAPA PCR mixture (12.5 $\mu$l of 2× KAPA HiFi HotStart ReadyMix, 0.2 $\mu$l of 10 $\mu$M IS PCR primer, 2.3 $\mu$l of DDW), and applying the thermal cycling program for HiFi.

### Protease-Smart-seq2 cDNA amplification

Cells in formalin-fixed sections were isolated with LCM in 6.4 $\mu$l of cell lysis buffer (0.6 $\mu$l of 5× SuperScript II buffer, 0.1 $\mu$l of 100 $\mu$M Oligo dT VN, 0.8 $\mu$l of dNTP mix [25 mM each], 0.25 $\mu$l of 40 U/$\mu$l recombinant RNase inhibitor, 0.5 $\mu$l of 100 mM DTT, 0.06 $\mu$l of 1 M MgCl$_2$, 2 $\mu$l of 5 M Betaine, 0.4 $\mu$l of 5% Triton X-100, 0.32 $\mu$l of 900 mAU/ml QIAGEN Protease, and 1.37 $\mu$l of DDW) in the caps of PCR tubes, and spun down into the tubes by brief centrifugation. The isolated cells were lysed by protease digestion at 50°C for 10 min and then heat inactivated at 80°C for 15 min, followed by the addition of 0.8 $\mu$l of 1:2,500,000 diluted ERCC RNA Spike-In Mix 1 and incubation at 72°C for 90 s. Then, 3.6 $\mu$l of a template-switching mixture (1.4 $\mu$l of 5× SuperScript II buffer, 0.1 $\mu$l of 100 $\mu$M N-template-switching oligo, 0.25 $\mu$l of SuperScript II, 0.25 $\mu$l of SuperScript III, 0.2 $\mu$l of recombinant RNase inhibitor, and 1.4 $\mu$l DDW) was added, followed by the cycling RT program for DRaqL-Smart-seq2. Then, cDNA amplification was performed by adding 15 $\mu$l of the SeqAmp PCR mixture, and applying the thermal cycling program for SeqAmp. The amplified cDNA (25.8 $\mu$l) was purified with a 0.8× volume of AxyPrep MAG PCR clean-up reagent according to the manufacturer's instructions.

### Pooled and bulk RNA-seq of granulosa cells using DRaqL-SC3-seq

For pooled cell analysis, about 10 granulosa cells were isolated with LCM from the regions neighboring oocytes and non-neighboring regions in early antral follicles in the alcohol-fixed, mouse-frozen ovarian sections (n = 7 each). Cell lysis and cDNA amplification were performed using the DRaqL-SC3-seq method with a reduced number of PCR cycles (18 cycles).

For bulk RNA-seq of granulosa, the whole granulosa of individual sectioned follicles were isolated using LCM without considering the histological affiliation of individual cells. Early antral follicles were identified in the sections that contained oocytes (n = 4), for which oocytes were first removed by LCM and then whole granulosa were collected. The developmental stages of follicles that did not contain oocytes within the investigated sections were not identified. Cell lysis and cDNA amplification were performed using the DRaqL-SC3-seq method with a reduced number of PCR cycles (18 cycles).

### Quantification of amplified cDNA

The amount of the amplified cDNAs was measured with Qubit 4 Fluorometer using 1X dsDNA High-Sensitivity Kit (Thermo Fisher Scientific) according to the manufacturer's instruction. Briefly, 1 $\mu$l cDNAs and 10 $\mu$l standard solutions were added to 199 $\mu$l and 190 $\mu$l of 1X working solution, respectively, in a 0.5 ml thin-wall, clear PCR tube (Axygen). The mixtures were vortex and incubated for 2 min at room temperature.

### Gene expression analysis with real-time PCR

Real-time PCR was performed using a CFX384 real-time PCR system (Bio-Rad) with Power SYBR Green PCR Master Mix (Thermo Fisher Scientific), according to the manufacturer's instructions. The primers used for real-time PCR were as previously described (Nakamura et al, 2015) and are listed in Table S5.

### Quality control of amplified cDNAs

The success rate of the cDNA amplification from single cells isolated with LCM was evaluated as follows. The expression level of a housekeeping gene, *Arbp*, was quantified with real-time PCR. Then, the cDNAs for which the Smirnov–Grubbs test for outliers yielded a *P*-value < 0.01 were considered to have been unsuccessfully amplified and were removed from the subsequent analyses (Fig S1). The size of the amplified cDNA was analyzed using Bioanalyzer with High-Sensitivity DNA kit (Agilent Technologies).

### Library preparation for 3′-sequencing with DRaqL-SC3-seq

Preparation of the libraries for SC3-seq was performed as previously described (Nakamura et al, 2015, 2017) with minor modifications. Before the library preparation, ~5 ng of amplified single-cell cDNAs were pre-amplified with thermal cycling program 3 in 15 $\mu$l V1-NV3 pre-amplification mixture (containing 1.5 $\mu$l of 10× ExTaq Buffer, 1.2 $\mu$l of 2.5 mM dNTP, 0.15 $\mu$l of 1 $\mu$g/$\mu$l N-V3[dT]$_{24}$, 0.15 $\mu$l of 1 $\mu$g/$\mu$l V1[dT]$_{24}$, and 0.075 $\mu$l of ExTaqHS). The pre-amplified cDNAs were subjected to a primer–dimer removal by 6–9 cycles of DNA

purification with a 0.6× volume of AxyPrep reagent, and elution in 20 $\mu$l DDW. The size-selected cDNAs were then fragmented with Picoruptor (Diagenode) with 10 cycles of a 30-s sonication and a 30-s interval at 4°C, followed by end-polishing in 25 $\mu$l End polish mix (20 $\mu$l of the sonicated DNA, 2.5 $\mu$l of 10× NEB Next End Repair Reaction Buffer [NEB], 0.085 $\mu$l of T4 polynucleotide kinase [NEB], 0.085 $\mu$l of T4 DNA polymerase [NEB], and 2.33 $\mu$l of DDW), and incubating at 20°C for 30 min. The fragmented, end-polished cDNAs were then subjected to size selection, with 0.7× volume AxyPrep reagent for cutting off the larger molecular size, and the supernatant was recovered by supplementing with an additional 0.2× volume of AxyPrep reagent (0.9× volume total) and eluted in an 8 $\mu$l DDW. Next, internal adapter extension was performed in a 10-$\mu$l internal-adapter mixture (1 $\mu$l of 10× ExTaq Buffer, 0.94 $\mu$l of 2.5 mM dNTP, 0.67 $\mu$l of 10 $\mu$M tRd2SPV1[dT]$_{20}$, 0.067 $\mu$l of ExTaqHS, and 7.3 $\mu$l of the eluted DNA) with thermal cycling program 4. Then, the T-adapter ligation was performed by adding 6.63 $\mu$l of a ligation mixture (3.3 $\mu$l of 5× NEB Next Quick Ligation Reaction Buffer [NEB], 0.2 $\mu$l of 10 $\mu$M tRd1SP adapter [double-stranded DNA formed by annealing tRd1SPTs and tRd1SPTas oligos], 0.33 $\mu$l of 2,000 U/$\mu$l T4 DNA ligase [NEB], and 2.8 $\mu$l of DDW), and incubating at 20°C for 15 min and 72°C for 20 min, and the ligation product was purified with a 0.8× volume of AxyPrep reagent. Finally, an additional PCR was performed in a 20-$\mu$l S5-N7 PCR mixture (2 $\mu$l of 10×ExTaq Buffer, 1.6 $\mu$l of 2.5 mM dNTP, 2 $\mu$l of 10 $\mu$M S5-index primer, 2 $\mu$l of 10 $\mu$M N7-index primer, 0.1 $\mu$l of ExTaqHS, 2.3 $\mu$l of DDW, and 10 $\mu$l of the eluted DNA) with thermal cycling program 5, followed by DNA purification with a 0.9× volume of AxyPrep reagent. Then, libraries with different index sequences were mixed and subjected to sequencing from the Read1 primer (75-bp) using a NextSeq 500/550 High-Output Kit v2.5 (75 cycles) (Illumina).

### Library preparation for Y-shaped adapter sequencing for cDNAs prepared with the DRaqL-SC3-seq cDNA amplification method

Single-cell cDNAs were pre-amplified in a 15 $\mu$l V1–V3 pre-amplification mixture (1.5 $\mu$l of 10×ExTaq Buffer, 1.2 $\mu$l of 2.5 mM dNTP, 0.15 $\mu$l of 1 $\mu$g/$\mu$l V1[dT]$_{24}$, 0.15 $\mu$l of 1 $\mu$g/$\mu$l V3[dT]$_{24}$, 0.075 $\mu$l of ExTaqHS, 7 $\mu$l of DDW, and 5 $\mu$l of the cDNA) with thermal cycling program 3. Pre-amplified cDNAs were then subjected to the primer–dimer removal, sonication, end-polishing, and size selection as in the 3′-sequencing with DRaqL-SC3-seq described above. Then, purified cDNA was dA-tailed in a 10-$\mu$l dA tailing mixture (1 $\mu$l of 10× ExTaq Buffer, 0.94 $\mu$l of 2.5 mM dNTP, 0.066 $\mu$l of ExTaqHS, 8 $\mu$l of the purified DNA) with incubation at 72°C for 10 min. The dA-tailed product (10 $\mu$l) was subjected to the Y-shaped adapter ligation by adding a 6.6-$\mu$l Y-adapter mixture (3.3 $\mu$l of 5×NEB Next Quick Ligation Reaction Buffer, 0.2 $\mu$l of 10 $\mu$M Y-shaped S5-index adapter [Y-shaped DNA formed by annealing rP7-Ycore and P5-S5-index oligos, 0.33 $\mu$l of 2,000 U/$\mu$l T4 DNA Ligase, and 2.8 $\mu$l of DDW) and incubating at 20°C for 15 min and at 72°C for 20 min. The ligation product was purified with a 0.9× volume of AxyPrep reagent, and subjected to additional PCR amplification in 20 $\mu$l of the S5-N7 PCR mixture with the thermal cycling program 5, followed by DNA purification with a 0.9× volume of AxyPrep reagent. The libraries were mixed and subjected to sequencing from the Read1 and Read2 primers (75-bp each) using a NextSeq 500/550 High-Output Kit, ver. 2.5 (150 Cycles) (Illumina). We applied 10 samples of single-cell cDNAs from each cell type to the

Y-shaped adapter sequencing, and used samples of the top 5 uniquely mapped reads (1.8–4 M reads) for subsequent analyses, to remove samples with small numbers of mapped reads (<1.5 M reads).

### Library preparation for DRaqL-SMART-Seq v4, DRaqL-smart-seq2 and DRaqL-Protease-Smart-seq2

Libraries of the cDNAs amplified with the Smart-seq2-based methods (DRaqL-Smart-seq v4, DRaqL-Smart-seq2 and DRaqL-Protease-Smart-seq2) were prepared using the library construction module of the SMART-Seq v4 3′ DE Kit, according to the manufacturer's instructions. Briefly, 10–12 purified cDNA samples were pooled and purified with a 0.8× volume of AxyPrep reagent. 300–400 pg cDNAs were tagmented using the Amplicon Tagment Mix of the Illumina Nextera XT DNA Sample Preparation Kit (Illumina). 3′-ends of the tagmented cDNAs were amplified with Nextera PCR Master Mix and dual index primers (Reverse PCR primer HT Index and Tn Forward PCR Primer HT Index in the SMART-Seq v4 3′ DE Kit) designed for TnRP1 and TnRP2 sequences. All libraries were mixed and sequenced using the NextSeq 500/550 High-Output Kit v2.5 (75 Cycles) with paired-end mode using Read2 (8 bps) for the in-line index and with dual-index mode (8 bps).

### Single-cell RNA-seq of mouse ovarian oocytes and granulosa cells with DRaqL-SC3seq

Single oocytes (72 cells) in the primary-to-early antral follicles and the single granulosa cells associated with these oocytes in secondary-to-early antral follicles (140 cells) were histologically identified in the frozen ovarian sections stained with cresyl violet, and isolated with LCM. cDNAs were amplified with DRaqL from the isolated single cells, and 45 oocytes and 94 granulosa cells were passed through the Smirnov–Grubbs test for outliers ($P > 0.01$). We removed an oocyte and granulosa cells with low mapping rates (<3% and <13%, respectively), and subjected 44 oocytes and 61 granulosa cells (105 samples in total) to the 3′-sequencing (Nakamura et al, 2015), resulting in ~2.2 M and ~3.2 M uniquely mapped reads per single oocyte and granulosa cell, respectively (Fig S5). One cell showed a small number of detectable protein-coding genes (3,138) and low correlation coefficients of gene expression levels with all other cells in this study (<0.3), and was removed from subsequent analyses. In addition, after the sequence data processing described below, granulosa cells that showed mixed expression profiles with oocytes (3/60: 5%) were excluded from subsequent analyses. Details of evaluation of the mixed profiles are described below. Therefore, a total of 44 oocytes and 57 granulosa cells were finally used for the 3′-sequencing analyses. To examine whether known splice isoforms in oogenesis were detected, we subjected the cDNAs of 10 oocytes to the Y-shaped adapter sequencing and applied the top 5 samples with respect to the number of mapped reads to the subsequent analyses according to the above-described criteria.

### Data processing and transcriptome analysis for 3′-sequencing

Data processing for 3′ sequencing was performed according to the previously reported method with minor modifications. FASTQ files

were generated using the bcl2fastq tool (v. 2.20.0.422) and were trimmed to remove polyA/T sequences, adapter sequences, and poor-quality reads, using fastp (v. 0.20.0) (Chen et al, 2018) and cutadapt (ver. 1.16) (Martin, 2011). Then, reads were aligned to the GRCm38 mouse genome assembly using the HiSAT2 tool (v. 2.1.0) (Kim et al, 2015). Best-matched reads on annotated genes were counted using HTSeq 0.10.0 (Anders et al, 2015; Putri et al, 2021 *Preprint*). Gene annotation was performed as described previously (Nakamura et al, 2015), and reads mapped within 10 kb from the 3′-ends of genes annotated in the Ensembl database were included in the transcript counts of the genes.

For transcriptome analyses, read counts RPM matched to each gene were converted to $\log_2$ values ($\log_2$ [RPM+1]). Protein-coding genes with at least one mapped read were considered to be detectable genes. The detection rate of a gene in a group of cells was defined as the frequency of samples wherein the gene was detected. The two proportion $z$-test (two-sided) was performed to analyze the statistical significance of the difference of detection rates. The $t$ test (two sided) for differences in average expression levels was performed using $\log_2$ (RPM+1) values. False discovery rates of multiple testing were calculated using the Benjamini–Hochberg method (Benjamini and Hochberg, 1995). DEGs in granulosa cells were identified by the following criteria: difference of averaged $\log_2$ (RPM+1) >1.8 (i.e., >3.5-fold), which was employed according to the previous study using the same cDNA amplification method (Kurimoto et al, 2008); $\log_2$ (RPM+1) >4 in at least one sample; $P$-value by $t$ test < 0.05 and FDR <0.05. Enrichment of gene ontology terms in DEGs was calculated using the DAVID tool (Huang da et al, 2009a; Huang da et al, 2009b). PCA was performed for genes showing $\log_2$ (RPM+1) >4 in at least one sample, using the prcmp tool in the R program suite (R Core Team, 2020) with default settings, respectively. 97% confidence interval ellipses were calculated using the stat_ellipse tool in the R program suite.

### Data processing and exon profiling for Y-shaped adapter sequencing

Sequence trimming and genomic mapping were performed as described above. Reads mapped to exons and exon–exon junctions were counted for their expression profiling by featureCounts (v2.0.1) (Liao et al, 2014) with -J -M -t exon -g gene_id options. Taking account of the size variations among exons, we quantified the expression levels of exons and exon–exon junctions using TMM values, rather than RPM. The read counts were normalized using EdgeR (3.36.0) (Robinson et al, 2010), and transformed to TMM for expression level analyses. For expression profiling of exon–exon junctions, only genes with detectable expression levels in the 3′-sequencing were used, and exons and their junctions showing >2 TMM values were defined as detectable. To analyze expression levels, TMMs were transformed to $\log_2$ values ($\log_2$ [TMM+1]). To define the number of exons for each gene, we calculated the detection rates of exons for every transcript from each protein-coding gene registered in the Ensembl database, among all samples, and selected the transcript that showed the largest detection rate. The sequencing statistics for mESCs and cells of mouse ovaries (oocytes and granulosa cells) analyzed in this study are shown in Tables S6 and S7, respectively.

### Data processing for DRaqL-Smart-Seq v4, DRaqL-Smart-seq2 and DRaqL-Protease-Smart-seq2

The binary base call sequence files were converted to fastq files by the bcl2fastq tool with –minimum-trimmed-read-length 0 –mask-short-adapter-reads 0 –no-lane-splitting options. The fastq files were demultiplexed by using a SMART-Seq DE3 Demultiplexer. Quality control, read mapping, and read counting were performed for Read1 sequences by the same method as the data processing of the DRaqL-SC3-seq datasets.

### Statistical model of the oocyte transcriptome based on diameter

To reconstruct the transcriptome of oocytes, we employed simple linear regression analyses. PC1 and PC2 of oocytes served as objective variables, whereas their corresponding diameter was taken as an explanatory variable. We calculated PC1 and PC2 for each diameter bin, ranging from 10 $\mu$m to 100 $\mu$m at 10-$\mu$m intervals. The gene expression level of each gene was then reconstructed by adding the products of the calculated PC values and factor loadings to the mean $\log_2$-expression level. This process allowed us to create the reconstructed transcriptome for oocyte corresponding to each diameter bin. Subsequently, we calculated the Spearman's rank correlation coefficient ($rs$) between the reconstructed transcriptomes and the transcriptome of each individual oocyte. The transcriptome with the highest $rs$ value was considered the best-matched reconstructed transcriptome for that specific oocyte. We annotated oocytes with diameters significantly different from those of their best-matched models (>20 $\mu$m difference). To assess the similarity between the reconstructed transcriptomes and transcriptomes of real oocytes, we selected the top 500 genes with the largest positive and negative contributions to PC2, and performed hierarchical clustering using the Ward.D2 method in the R program suite.

### Estimation of RNA copy number

The mRNA copy number was estimated for each gene using the spike-in RNAs (ERCC). The read number of each gene was fitted to the linear regression model of $\log_2$-transformed read numbers and known copy numbers of ERCC RNAs, using the lm tool of the R program suite. The read numbers of the ERCC RNAs were obtained from the mapped sequence data using the idxstats function in samtools. ERCC RNAs that showed >0 reads were used for the calculation of linear regression models.

### Single-cell transcriptome analysis of freshly dissociated oocytes and granulosa cells, and those isolated from frozen sections through LCM, from ovaries in the same mouse

The estrous cycle of an 8-wk-old mouse was determined using vaginal smear analysis. The mouse was euthanized during proestrus, and its ovaries were isolated in L15 medium (L5520; Sigma-Aldrich). One ovary was immediately frozen in liquid $N_2$ for subsequent LCM-based single-cell transcriptome analysis using DRaqL-Smart-seq2.

From the other ovary, COCs were isolated from fully grown follicles using tweezers and a needle into L15 medium containing 240 $\mu$M Dibutyryl-cAMP (#D0627; Sigma-Aldrich), an analog of cAMP. This COC isolation was conducted as previously reported (Takashima et al, 2021), with a minor modification in that natural estrus cycle was investigated without superovulation. The oocytes and their attached granulosa cells, and other granulosa cells within COCs, were dissociated by pipetting and then isolated using a glass capillary under visual inspection with a stereomicroscope. Each cell was washed twice in 200 $\mu$l of a 0.1% PVA solution in PBS. These freshly dissociated oocytes and granulosa cells were collected into cell lysis buffer, and cDNAs were amplified using Smart-seq2.

RNA sequencing and data processing were performed using the same method as described above, with slight modifications in the quality control process: granulosa cells with less than 100,000 reads and <3,000 detectable protein-coding genes, and oocytes with less than 100,000 reads and <10,000 detectable protein-coding genes, were excluded from further analysis. Consequently, a total of 14 out of 16 freshly dissociated oocytes, 8 out of 8 LCM-isolated oocytes from frozen sections, 35 out of 80 freshly dissociated granulosa cells, and 70 out of 80 LCM-isolated granulosa cells from frozen sections were included in the subsequent analysis.

### Evaluation of mixed expression profiles of oocytes and granulosa cells

To assess the influence of oocyte transcriptome on adjacent granulosa cells in frozen sections analyzed using DRaqL-SC3-seq, we compared transcriptome of granulosa cells with the average expression profile of oocytes using scatterplots of $\log_2$ gene expression levels with contour profiles. We observed a subset of granulosa cells (5%, 3/60) displaying bimodal patterns in the contour plots, indicating the presence of mixed expression profiles of granulosa cells and oocytes (see Fig S7A and B).

To further investigate the existence of mixed expression profiles, we utilized the single-cell RNA-seq data of freshly dissociated oocytes and granulosa cells described above, ensuring the analysis of genuine single cells without contamination. Consistent with the findings from the frozen sections, the contour plots of $\log_2$ expression levels between granulosa cells and oocytes also exhibited bimodal patterns (Fig S8). Notably, granulosa cells directly attached to oocytes showed higher frequency of these mixed profiles (42% [6/14]) compared with other granulosa cells (5% [1/21]). Thus, we consider that these profiles are authentic expression profiles rather than a result of contamination. However, to ensure a focused investigation of the pure transcriptome of granulosa cells, we excluded the cells exhibiting mixed profiles from subsequent analyses using DRaqL-SC3-seq, resulting in the detailed exploration of the remaining 57 granulosa cells (Figs 4, 5, and 6).

### Analysis of published datasets for oocytes and granulosa cells

A previously published single-cell RNA-seq dataset for mouse ovarian somatic cells (CRA003928) (Li et al, 2021) was downloaded and re-analyzed using the Scanpy tool (Wolf et al, 2018) in the Seurat v4.1.1 program suite (Hao et al, 2021) with the previously reported settings (Li et al, 2021). The features were filtered as follows: 200 < nFeature < 5,000, 1 < percent.mt < 5 and nCount_RNA > 1,000. The filtered feature counts were normalized by the total number of tags, and the counts per 10,000 reads were calculated using the NormalizedData function with LogNormalize method. Highly variable genes for subsequent analyses (6,883 genes) were identified by the FindVariableFeatures tool with the dispersion method and variable nfeatures. UMAP was calculated using the RunUMAP tool with dims = 1:50, and identified 12 clusters, which were then annotated using the marker genes as reported previously (Li et al, 2021). Cluster 1 was identified as cumulus cells based on the expression of *Fabp5* and *Ldha*, and low/no detectable expression of corpus luteal marker (*Hsd3b1*, *Inhba*) and early granulosa markers in pre-antral follicles (*Birc5*, *Cdc8*). Among all genes annotated, the DEGs between the neighboring and non-neighboring granulosa cells identified in the DRaqL-SC3-seq analysis (Fig S16) were extracted for further analysis. The average expression levels of the genes up-regulated in the neighboring and non-neighboring cells were calculated, and cells were ordered according to the rank of the difference between these expression levels. The expression level differences between the top and bottom 25% of cells were examined with the Wilcoxon rank test.

For the analysis of single-cell RNA-seq of human oocytes and granulosa cells (GSE186504) (Fan et al, 2021), expression data were downloaded, and 130 granulosa cells were identified based on the information in SRA Run Selector and their gene expression profiles. The average expression levels (RPM values) of the above DEGs were calculated and ordered according to the rank of the expression level difference followed by the Wilcoxon rank test as described above.

For the comparison with transcriptome data during mouse oogenesis from nongrowing to germinal vesicle (GV) oocytes (Gahurova et al, 2017), we downloaded RNA-seq read count data from GSE86297 and calculated $\log_2$ (sample read counts per million reads + 1) values. To identify DEGs among nongrowing and growing oocytes, we performed analysis of valiance and identified 1,372 genes that exhibited $P < 0.05$, a minimum $\log_2$ expression level >4 in at least one sample, and a minimum $\log_2$ expression level difference > 1 in at least one pair of samples. Among these genes, we identified 1,286 whose gene symbols were found in the gene table used in our study (GRCm39). For these genes, we calculated correlation coefficients with our transcriptome data of single oocytes.

For evaluation of the activation phase of granulosa cells, we investigated genes identified by Morris et al (2022) (Supplementary File 2: top 10 markers expressed in each ovary cluster) (Morris et al, 2022). We analyzed $\log_2$ RPM+1 values in the single granulosa cells and oocytes in our dataset for "Granulosa," "Preantral-Cumulus," "Antral-Mural," "Atretic," "Mitotic," "Luteinizing mural," "Active CL," "Mitotic-Antral," and "Regressing CL" (CL, corpus luteum).

For the comparison with transcriptome data of human oocytes published by Ernst et al (2017), we downloaded data of DEGs between oocytes in primordial and primary follicles (Table S1 in Ernst et al [2017]: DEGs). Human Ensembl gene IDs were converted to mouse gene ID using g:Profiler (Raudvere et al, 2019). 249 and 186 genes were identified as up- and down-regulated in human primary-follicle oocytes, respectively, sharing orthologues with mice. Z scores were calculated using the primary oocyte mean (FPKM) values of these genes. We also calculated Z scores for average $\log_2$ RPM+1 values in the primary-follicle oocytes in our

dataset for mouse orthologues. These Z scores were compared between human and mouse, and genes that exhibited more than twofold differences were identified as differentially expressed. Gene ontology analysis was performed for DEGs between humans and mice using the DAVID tool.

## Immunofluorescence

Snap-frozen mouse ovary embedded in optimal cutting temperature compound (Tissue-Tek) were sectioned with a 6-$\mu$m thickness and mounted on MAS-GP typeA glass slides (MATSUNAMI). Ovarian sections were fixed with 50% iso-propanol for 30 s at RT followed by permeabilization in 1×PBS containing 1% Triton X-100 (12967-32; nacalai tesque) for 10 min and three times of 5-min washing in 1×PBS. Then, the ovarian sections were incubated with 5% BSA (Sigma-Aldrich) and 10% in 1×PBS donkey serum (Sigma-Aldrich) for 60 min at RT for blocking. Then, sections were incubated with primary antibodies in 1×PBS containing 1% BSA in a dark humidity chamber for 1 h at RT, followed by three 5-min washes in 1×PBS. The following primary antibodies were used at a dilution rate of 1:00 in this study: rabbit anti-PBX1 antibody (18204–1-AP; Proteintech): rabbit anti-SUZ12 antibody (3737T; Cell Signaling Technology). Then, the sections were incubated with secondary antibody (donkey anti-rabbit Alexa Fluor 594 #A21207; Life Technologies) at a dilution rate of 1:300 in 1× PBS containing 1% BSA for 1 h at RT, followed by three 5-min washes in 1×PBS, and were mounted in VECTASHIELD Vibrance Antifade Mounting Medium with DAPI (VECTOR LABORA-TORIES) with a cover glass (MATSUNAMI).

These ovarian sections were analyzed with confocal laser microscopy using Olympus FV3000 with a ×20 objective lens. For quantitative analysis of fluorescent signals, regions of nuclei were identified with DAPI signals using ImageJ ROI manager, and signal densities of the target proteins in nuclei were quantified using the defined ROIs. Four follicles were analyzed for the measurement of fluorescence signal intensities for both proteins. Robust z-score of the signal intensities were calculated and *Wilcoxon* test were performed by the *R* program suit.

# Data Availability

All raw and processed sequencing data generated in this study have been submitted to the NCBI Gene Expression Omnibus (GEO; https://www.ncbi.nlm.nih.gov/geo/) under accession number GSE192551.

# Supplementary Information

# Acknowledgements

We thank all the members of our laboratory for their discussion and advice regarding this study, and Junko Komeda for secretarial support. We also thank Kazusa Ohkita and the Single-Cell Genome Information Analysis Core (SignAC) in ASHBi for the RNA sequence analysis. Finally, we thank Doctors Horvath and Migh for their kind advice about the experimental settings for LCM. This study was supported by KAKENHI grants (JP20H00471, JP18K19295, and JP18H05553) to K Kurimoto; by funds from the Uehara Memorial Foundation, the Takeda Science Foundation, and the Daiichi Sankyo Foundation of Life Science to K Kurimoto; and by a KAKENHI grant (JP21K15107) to H Ikeda.

## Author Contributions

H Ikeda: conceptualization, data curation, formal analysis, funding acquisition, investigation, visualization, methodology, and writing—original draft.
S Miyao: methodology.
S Nagaoka: conceptualization and writing—review and editing.
T Takashima: investigation.
S-M Law: investigation.
T Yamamoto: investigation.
K Kurimoto: conceptualization, data curation, formal analysis, supervision, funding acquisition, investigation, visualization, project administration, and writing—original draft, review, and editing.

## Conflict of Interest Statement

H Ikeda, S Miyao, and K Kurimoto have a patent pending in Japan (application number 2021-200053) on this method. K Kurimoto has a patent for the cDNA amplification method used in this study (patent number USA 9222129, Japan 5526326).

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
