## [Reviewer comments · Life Science Alliance]

Life Science Alliance

High-quality single-cell transcriptomics from ovarian histological sections during folliculogenesis

Hiroki Ikeda, Shintaro Miyao, So Nagaoka, Tomoya Takashima, Sze-Ming Law, Takuya Yamamoto, and Kazuki Kurimoto
DOI: <https://doi.org/10.26508/lsa.202301929>

Corresponding author(s): Kazuki Kurimoto, Nara Medical University

Review Timeline:

Submission Date:	2023-01-16
Editorial Decision:	2023-03-20
Revision Received:	2023-07-26
Editorial Decision:	2023-08-21
Revision Received:	2023-08-23
Accepted:	2023-08-23

Scientific Editor: Novella Guidi

Transaction Report:

March 20, 2023

Re: Life Science Alliance manuscript #LSA-2023-01929-T

Kazuki Kurimoto
School of Medicine, Nara Medical University

Dear Dr. Kurimoto,

Thank you for submitting your manuscript entitled "Histology-associated transcriptomic heterogeneity in ovarian folliculogenesis revealed by quantitative single-cell RNA-sequencing for tissue sections with DRaQL" to Life Science Alliance. The manuscript was assessed by expert reviewers, whose comments are appended to this letter. We invite you to submit a revised manuscript addressing the Reviewer comments.

Thank you for this interesting contribution to Life Science Alliance. We are looking forward to receiving your revised manuscript.

Sincerely,

B. MANUSCRIPT ORGANIZATION AND FORMATTING:

Reviewer #1 (Comments to the Authors (Required)):

The manuscript "Histology-associated transcriptomic heterogeneity in ovarian folliculogenesis revealed by quantitative single-cell RNA-sequencing for tissue sections with DRaQL" provides advances in methodology to allow single cell transcriptional profiling of single cells isolated from histological sections containing dissociated stem cells, that are easily captured by laser capture microscopy as individual cells.

However, we usually don't need to embed dissociated cells, but rather tissue that is much more challenging to capture by laser capture microscopy as individual cells.

The authors further test their technology on oocyte and granulosa cells from histological sections of mouse adult ovaries.

There are several points that need addressing before further consideration:

- what is the difference between this technology and the one applied by Ernst et al 2017 (doi:10.1093/humrep/dex238)? And how do the profiles/signatures of oocyte and granulosa cells between the two studies differ? Perhaps considering major differences and similarities between humans and mice during follicular activation and associated oocyte growth.
- How do the authors make sure that laser capture microscopy only captures one (granulosa) cell? How are doublets or granulosa cells contaminated with other cells removed from the data? What percentage of the data contains more than one single cell or contamination from different cell types? How can this be corrected for? For this we need careful analysis of the obtained transcriptomics and comparison with conventional single cell transcriptomics data isolated from the same material, regarding mixed signatures. In addition, more detail about the single-cell capture procedure should be provided.
- if I understand correctly, the authors do not directly compare ovary from the same donor (in the same ovarian phase, age, etc) using their technology and smartseq. Instead, they compare with existing datasets from human and mouse. Although this is important, to understand the advantages of the developed technology the authors would need to use the same material to compare between their newly developed technology and the regular smartseq using the same material directly and compare the signatures.
- It is unclear what the biology means: in what stage of activation are the granulosa cells and the oocytes? Just mentioning correlation with size without mentioning quantitative markers associated with the different markers that could correlate with different phases of oocyte and granulosa activation it is difficult to understand the utility of the data. The analysis needs to be more correlated with the biology (for example compare with Ernst et al 2017).
- the authors do not mention atresia: does it mean that none of the oocytes/follicles analysed were undergoing atresia? This seems highly unlikely.
- None of the data provided, but particularly the 'newly identified co-expression markers', are validated in sections of mouse ovaries: are there novel markers or features identified that could be validated? This would increase the robustness of the paper.

Reviewer #2 (Comments to the Authors (Required)):

Ikeda et al. reported a study to optimize laser-capture-microdissection single-cell RNA sequencing method for ovarian tissues. The authors tested many conditions including fixation, lysis of cells, cDNA construction and etc. After optimization with cell block derived from cell culture, they used mouse ovarian sections to test the efficiency of the developed method. They described the correlation of transcriptomes with the diameter of the captured follicles and the correlation of relative location of the granulosa cells with their transcriptomes. Overall, the study presented detailed and comprehensive methodology development of LCM single-cell transcriptome for ovarian tissue, potentially applicable to other tissues.

The results are presented logically and the conclusions are mostly supported by the datasets. However, there are results which needs more clarifications, for instance, in page 23, line 9-14. 'We found antagonistic expression patterns of these DEGs in subsets 12 of granulosa cells in the public datasets (Supplemental Fig. S11).' It is not clear which DEGs and antagonistic expression patterns the authors referred to.

Rebuttal (LSA-2023-01929-TR)

We would like to express our gratitude to the reviewers for their valuable time and effort in providing constructive feedback. In response to their comments, we have made revisions to our manuscript, including the addition of new experiments and analyses as follows:

- (1) We included a new dataset of single-cell RNA-seq from both freshly dissociated cells and LCM-isolated cells in ovarian sections from the same mouse to analyze the mixed expression profiles of granulosa cells and oocytes and to verify the robustness of our method.
- (2) We performed immunofluorescence on granulosa cells to verify the histology-associated expression patterns of granulosa cells.
- (3) We compared our methods with those used in a previous study (Ernst et al., 2017), in addition to performing a comparative analysis of human and mouse oocytes.
- (4) We have incorporated results from previous studies on gene expression dynamics in oocytes and granulosa cells to clarify the biological relevance of our expression data and their association with oocyte morphology.

As a result, we have added seven additional Supplementary Figures in the revised version of the manuscript (Supplementary Figs S6–S8, S13, S14, S17, and S18). Additionally, we have adjusted the manuscript format to align with the guidelines, and we have integrated the contents of Supplemental Materials and Methods in the original version into main text for better coherence. The revised sections are indicated with red letters to make it easier to identify the changes. We believe that these revisions have significantly improved our manuscript and that it is now worthy of publication in *Life Science Alliance*. Here, we address each of the reviewer's comments in detail as follows.

Reviewer #1 (Comments to the Authors (Required)):

Comment 1

The manuscript "Histology-associated transcriptomic heterogeneity in ovarian folliculogenesis revealed by quantitative single-cell RNA-sequencing for tissue sections with DRaqL" provides advances in methodology to allow single cell transcriptional profiling of single cells isolated from histological sections containing dissociated stem cells, that are easily captured by laser capture microscopy as individual cells.

However, we usually don't need to embed dissociated cells, but rather tissue that is much more challenging to capture by laser capture microscopy as individual cells.

The authors further test their technology on oocyte and granulosa cells from histological sections of mouse adult ovaries.

Response 1

We appreciate the reviewer's recognition of the methodological advancements presented in our study.

Comment 2

There are several points that need addressing before further consideration:

- what is the difference between this technology and the one applied by Ernst et al 2017 (doi:10.1093/humrep/dex238)?

Response 2

We thank the reviewer for this comment on the technical characteristics of our methods. The methods developed in our study differ from that used in the previous study by Ernst et al. (Ernst et al. 2017) in the following ways. First, our methods enable quantitative transcriptome analysis of single oocytes and granulosa cells isolated from frozen and formalin-fixed sections, while the method used in the previous study allowed RNA-seq analysis of pooled oocytes (45–186 oocytes) that were isolated from ovarian sections. Secondly, our methods amplify cDNAs directly from single cells isolated from tissue sections without the need for RNA purification, which was achieved through the sequential use of denaturing and non-denaturing detergents. In contrast, the method used in the previous study extracts and purifies RNA from pooled oocytes to subject them to cDNA amplification. As a result, our methods are significantly more sensitive and less labor-intensive than those used in the previous study. These points have been clarified in the revised manuscript (page 23, lines 4-8).

Comment 3

And how do the profiles/signatures of oocyte and granulosa cells between the two studies differ? Perhaps considering major differences and similarities between humans and mice during follicular activation and associated oocyte growth.

Response 3

In the previous study, Ernst et al. (Ernst et al. 2017) analyzed the transcriptome of human

oocytes in primordial to primary follicles but did not include granulosa cells, while in our study, we analyzed oocytes in primary to early antral follicles and granulosa cells in mice. As a result, we were able to compare the oocyte profiles in primary follicles in humans and mice between the previous study and ours (Fig S18 in the revised manuscript). We focused on genes up- and down-regulated during human primordial-to-primary follicle transition. In this analysis, we revealed that genes related to mTOR signaling and GnRH signaling pathways, featured in the previous study, were differentially expressed between human and mouse oocytes, while a majority of the genes were expressed at comparable levels. In addition, FOXO1, a key transcription factor expressed in human oocytes, also showed different expression in mouse oocytes. These data may highlight species difference in the molecular mechanisms of oocyte activation, and have been discussed in the revised version of manuscript (page 23, lines 8-15; page 43, line 27 – page 44, line 10).

Comment 4

- How do the authors make sure that laser capture microscopy only captures one (granulosa) cell? How are doublets or granulosa cells contaminated with other cells removed from the data? What percentage of the data contains more than one single cell or contamination from different cell types? How can this be corrected for? For this we need careful analysis of the obtained transcriptomics and comparison with conventional single cell transcriptomics data isolated from the same material, regarding mixed signatures. In addition, more detail about the single-cell capture procedure should be provided.

Response 4

We thank the reviewer for this comment on the principle and limitation of our methods. First, we excluded doublets of granulosa cells through careful microscopic inspection of ovarian sections. Because the thickness of sections was 15 μm , which is similar to the size of a single granulosa cell, it is unlikely that two or more cells were isolated with this procedure. We have also provided more details about the procedure for our single-cell capture in the revised manuscript (Fig S6; page 26, line 18 – page 27, line 4). However, we cannot completely rule out the possibility of contamination from adjacent granulosa cells. We have clarified this point in the revised version (page 21, lines 11-12).

Secondly, during the data processing conducted in our original manuscript, we indeed identified a subset of granulosa cells (5%, 3/60) that exhibited expression profiles mixed

with those of oocytes in a transcriptome-wide manner. In contour plots, the gene expression comparison between these granulosa cells and average of oocytes exhibited bimodal patterns, which were most likely attributable to the mixture of the distinct expression patterns of oocytes and granulosa cells (Fig S7 in the revised manuscript). As described in the Supplemental Materials and Methods of the original manuscript we excluded these cells from our analysis. (See the original Supplemental Materials and Methods section: “*In addition, after the sequence data processing described below, granulosa cells that showed mixed expression profiles with oocytes were excluded from subsequent analyses*”).

To further examine the existence of the mixed expression profiles, we performed single-cell RNA-seq analysis on freshly dissociated oocytes and granulosa cells from antral follicles from one of the ovaries of a female mouse, using the conventional Smart-seq2 method. These cells were isolated under stereomicroscopic inspection, including granulosa cells directly attached to oocytes, and thus were considered to be bona fide single cells without contamination from other cells.

Among these freshly dissociated cells, we also observed granulosa cells that showed bimodal patterns in the contour plots of gene expression comparison with oocytes, evidencing the presence of mixed expression profiles (Fig S8). Furthermore, these mixed profiles were more frequently found in granulosa cells attached to oocytes than in other granulosa cells (43% [6/14] versus 5% [1/21], respectively) (page 14, line 26 – page 15, line 4).

In addition, in a previous single-cell RNA-seq dataset of freshly dissociated human oocytes and granulosa cells, we observed that one of the granulosa cells displayed an expression profile highly similar to the oocyte transcriptome, suggesting existence of a transcriptome mix of oocytes and granulosa cells in human as well (Fig S16E; page 20, line 13–18).

These results suggest that the mixed expression profiles identified in our original manuscript may be a genuine mixture of granulosa cells and oocytes rather than resulting from contamination during LCM. These mixed profiles might be a result of cytoplasmic connections between oocytes and granulosa cells, such as transzonal projections, but investigating this possibility would be beyond the scope of this study.

In the revised version of our manuscript, to focus the analysis of pure transcriptome of granulosa cells, we again excluded the mixed profiles from the in-depth transcriptome analysis (Figs 4 and 6). We have clarified these points in the revised manuscript (page 10, line 27–page 11, line 6; page 19, lines 11-13; page 41, lines 19 – page 42, line 10).

Comment 5

- if I understand correctly, the authors do not directly compare ovary from the same donor (in the same ovarian phase, age, etc) using their technology and smartseq. Instead, they compare with existing datasets from human and mouse. Although this is important, to understand the advantages of the developed technology the authors would need to use the same material to compare between their newly developed technology and the regular smartseq using the same material directly and compare the signatures.

Response 5

We thank the reviewer for this critical point. However, in the original version of the manuscript, we used the same ovary for the comparison of all cDNA amplification methods developed in our study (DRaQL-SC3-seq, DRaQL-Smart-seq, DRaQL-SMART-seq v4, and DRaQL-Protease-Smart-seq2) and conventional single-cell RNA-seq methods (the regular Smart-seq [Smart-seq2], SMART-Seq v4, and SC3-seq). We believe that this strategy would provide the most robust evaluation for the performance of our methods. We have clarified this point in the revised manuscript (page 13, lines 22–25).

Additionally, in the revised manuscript, we performed DRaQL-Smart-seq2 to obtain transcriptome data from single granulosa cells and oocytes isolated using LCM from frozen ovarian sections. The ovary used in this RNA-seq analysis was obtained from the same mouse that was used for the conventional Smart-seq2 to obtain single cell transcriptome of freshly dissociated oocytes and granulosa cells, as described in Response 4 (Fig S8). These cells exhibited similar sensitivity and expression profiles, demonstrating the robustness of our method. We have clarified these points in the revised manuscript (page 13, line 27 – page 14, line 24).

Comment 6

- It is unclear what the biology means: in what stage of activation are the granulosa cells and the oocytes? Just mentioning correlation with size without mentioning quantitative markers associated with the different markers that could correlate with different phases of oocyte and granulosa activation it is difficult to understand the utility of the data. The

analysis needs to be more correlated with the biology (for example compare with Ernst et al 2017).

Response 6:

We thank the reviewer for this constructive comment. First, to clarify the stage of oocyte activation, we incorporated a previous RNA-seq dataset on non-growing to germinal vesicle (GV) oocytes in mice (GSE86297) (Gahurova et al. 2017). Using this dataset, we identified 1,286 genes exhibiting dynamic expression profiles during the oocyte growth (Fig S13A, S13B). Among these genes, 559 were highly expressed in non-growing oocytes and consistently decreased during oocyte growth. This subset included essential transcription factors for oocyte differentiation (*Figla*, *Sohlh1*, *Sohlh2*) and meiotic genes (*Sycp1*, *Sycp3*, *Smc1b*, *Syce1*). Additionally, 727 genes displayed a consistent increase in the growing oocytes, reaching their maximum expression levels in GV oocytes, and encompassed oocyte-specific transcription factors (*Obox1*, *Obox2*), a crucial signaling molecule for oogenesis (*Bmp5*), members of the Oogenesin family (*Oog1*, *Oog2*, *Oog3*, *Oog4*), and the DNA methyltransferase essential for the generation of the oocyte epigenome (*Dnmt3l*). Then, we calculated correlation coefficient of these genes between the previous dataset and our own (Fig S13C). We found that the primary-follicle oocytes in our study showed the highest degree of similarity to growing oocytes in the previous study, while oocytes from secondary-to-early antral follicles showed similarity to growing and GV oocytes in accordance to their respective diameter. Remarkably, the oocytes best matched to the reconstructed transcriptome for smaller diameter than their actual size, with a decrease of $>20\ \mu\text{m}$ (Fig 5D), displayed expression profiles similar to those of non-growing oocytes (Fig S13C). This suggests that these oocytes experienced growth retardation regarding to their transcriptome despite their larger size. These data clearly demonstrate that the size–transcriptome relationship of oocytes represents their growth phase. These points have been clarified in the revised manuscript (page 16, line 11 – page 17, line 7; page 43, lines 11-19).

Second, to examine the phase of granulosa activation, we incorporated a list of signature genes in the mouse ovary during the estrus cycle (Morris et al. 2022) (Fig S14). We found that both neighboring and non-neighboring granulosa cells showed high expression of granulosa-cell markers and signatures for mitotic granulosa cells in antral follicles and heterogeneous expression of preantral cumulus cells. In contrast, these cells showed very low or no expression of signature genes of mural granulosa cells, atretic follicles, and corpus luteum cells. These data indicate that the cells analyzed in our study were

mitotically active granulosa cells in preantral-to-antral follicles, consistent with our histological inspection. These points have been clarified in the revised manuscript (page 18, lines 11–17; page 43, lines 21–25).

Comment 7

- the authors do not mention atresia: does it mean that none of the oocytes/follicles analysed were undergoing atresia? This seems highly unlikely.

Response 7

As the reviewer noted, our data did not contain atretic cells, as demonstrated by the low-level expression or absence of the atretic markers (Fig S14). We believe that this is because we selected morphologically normal follicles through histological analysis before isolating cells for transcriptomic analysis. We have clarified this in the revised manuscript (page 18, lines 15–17; page 26, lines 18-20).

Comment 8

- None of the data provided, but particularly the 'newly identified co-expression markers', are validated in sections of mouse ovaries: are there novel markers or features identified that could be validated? This would increase the robustness of the paper.

Response 8

We thank the reviewer for this constructive comment. To address this issue, we performed immunofluorescence for PBX1, a homeobox transcription factor, and SUZ12, an epigenetic regulator contained in the Polycomb repressive complex. The mRNAs of both genes were up-regulated in the non-neighboring granulosa cells, as depicted in Fig 6 of our original manuscript. To the best of our knowledge, this type of differential expression of these genes is a novel finding. Consistent with the mRNA expression patterns of these genes, we found that PBX1 and SUZ12 exhibited significantly higher protein levels in non-neighboring granulosa cells than in neighboring cells in antral follicles, as shown in Fig S17 in the revised manuscript. These data validate the gene expression patterns associated with the histological information of granulosa cells in mouse ovarian sections. These points have been clarified in the revised version (page 20, lines 20 –page 21, lines 2; page 44, line 12 – page 45, line 7).

Reviewer #2 (Comments to the Authors (Required)):

Comment 1

Ikeda et al. reported a study to optimize laser-capture-microdissection single-cell RNA sequencing method for ovarian tissues. The authors tested many conditions including fixation, lysis of cells, cDNA construction and etc. After optimization with cell block derived from cell culture, they used mouse ovarian sections to test the efficiency of the developed method. They described the correlation of transcriptomes with the diameter of the captured follicles and the correlation of relative location of the granulosa cells with their transcriptomes. Overall, the study presented detailed and comprehensive methodology development of LCM single-cell transcriptome for ovarian tissue, potentially applicable to other tissues.

Response 1

We thank the reviewer for this encouraging comment.

Comment 2

The results are presented logically and the conclusions are mostly supported by the datasets. However, there are results which needs more clarifications, for instance, in page 23, line 9-14. 'We found antagonistic expression patterns of these DEGs in subsets 12 of granulosa cells in the public datasets (Supplementary Fig. S11).' It is not clear which DEGs and antagonistic expression patterns the authors referred to.

Response 2

We thank the reviewer for raising this question and allowing us to provide further clarification on the results of our analysis. The DEGs referred to in the analysis mentioned by the reviewer were specifically the genes that exhibited differential expression between neighboring and non-neighboring granulosa cells, as depicted in Fig 6B of our manuscript. Among these DEGs, 35 genes were up-regulated in the neighboring granulosa cells, while 97 genes were up-regulated in the non-neighboring granulosa cells (Table S6).

In the context of the previous single-cell RNA-seq studies conducted by Fan et al. (Fan et al. 2021) and Li et al. (Li et al. 2021), we were able to identify two groups of granulosa cells, each representing the top and bottom 25% based on the expression level difference between these DEG groups. Notably, the up-regulated genes in neighboring and non-neighboring cells were also differentially expressed between these top and bottom 25% cell groups with statistical significance ($p < 0.001$ with *Wilcoxon* test) (Fig S16A–S16D). These results demonstrate that DRaQL-SC3-seq for ovarian sections revealed previously unidentified co-expression patterns in

granulosa cells. To provide clarity on these findings, we have included a detailed explanation in the revised manuscript (page 20, lines 1–11; page 42, line 12 – page 43, line 9).

REFERENCES

- Ernst EH, Grondahl ML, Grund S, Hardy K, Heuck A, Sunde L, Franks S, Andersen CY, Villesen P, Lykke-Hartmann K. 2017. Dormancy and activation of human oocytes from primordial and primary follicles: molecular clues to oocyte regulation. *Hum Reprod* **32**: 1684-1700.
- Fan X, Moustakas I, Bialecka M, Del Valle JS, Overeem AW, Louwe LA, Pilgram GSK, van der Westerlaken LAJ, Mei H, Chuva de Sousa Lopes SM. 2021. Single-Cell Transcriptomics Analysis of Human Small Antral Follicles. *Int J Mol Sci* **22**.
- Gahurova L, Tomizawa SI, Smallwood SA, Stewart-Morgan KR, Saadeh H, Kim J, Andrews SR, Chen T, Kelsey G. 2017. Transcription and chromatin determinants of de novo DNA methylation timing in oocytes. *Epigenetics Chromatin* **10**: 25.
- Li S, Chen LN, Zhu HJ, Feng X, Xie FY, Luo SM, Ou XH, Ma JY. 2021. Single-cell RNA sequencing analysis of mouse follicular somatic cells. *Biol Reprod* doi:10.1093/biolre/ioab163.
- Morris ME, Meinsohn MC, Chauvin M, Saatcioglu HD, Kashiwagi A, Sicher NA, Nguyen N, Yuan S, Stavely R, Hyun M et al. 2022. A single-cell atlas of the cycling murine ovary. *Elife* **11**.

August 21, 2023

RE: Life Science Alliance Manuscript #LSA-2023-01929-TR

Prof. Kazuki Kurimoto
Nara Medical University
Department of Embryology
840 Shijo-Cho
Kashihara, Nara 634-8521
Japan

Dear Dr. Kurimoto,

Thank you for submitting your revised manuscript entitled "High-quality single-cell transcriptomics from ovarian histological sections during folliculogenesis". We would be happy to publish your paper in Life Science Alliance pending final revisions necessary to meet our formatting guidelines.

-please add a callout for Fig 3E, Fig S6A-B, Fig S7A, Fig S9A-B, Fig S14A-F, Fig S15A-H, Fig S17A-C, Fig S18A to your main manuscript text

A. FINAL FILES:

B. MANUSCRIPT ORGANIZATION AND FORMATTING:

Sincerely,

Reviewer #1 (Comments to the Authors (Required)):

Dear authors,
You have followed my suggestions and the manuscript has improved in quality considerably.

August 23, 2023

RE: Life Science Alliance Manuscript #LSA-2023-01929-TRR

Prof. Kazuki Kurimoto
Nara Medical University
Department of Embryology
840 Shijo-Cho
Kashihara, Nara 634-8521
Japan

Dear Dr. Kurimoto,

Thank you for submitting your Methods entitled "High-quality single-cell transcriptomics from ovarian histological sections during folliculogenesis". It is a pleasure to let you know that your manuscript is now accepted for publication in Life Science Alliance. Congratulations on this interesting work.

DISTRIBUTION OF MATERIALS:

Again, congratulations on a very nice paper. I hope you found the review process to be constructive and are pleased with how the manuscript was handled editorially. We look forward to future exciting submissions from your lab.

Sincerely,
